# Benefits of Assistance over Reward Learning

## Abstract

Much recent work has focused on how an agent can learn what to do from human feedback, leading to two major paradigms. The first paradigm is *reward learning*, in which the agent learns a reward model through human feedback that is provided externally from the environment. The second is *assistance*, in which the human is modeled as a part of the environment, and the true reward function is modeled as a latent variable in the environment that the agent may make inferences about. The key difference between the two paradigms is that in the reward learning paradigm, by construction there is a separation between reward learning and control using the learned reward. In contrast, in assistance these functions are performed as needed by a single policy. By merging reward learning and control, assistive agents can reason about the impact of control actions on reward learning, leading to several advantages over agents based on reward learning. We illustrate these advantages in simple environments by showing desirable qualitative behaviors of assistive agents that cannot be found by agents based on reward learning.

## 1 Introduction

Traditional computer programs are instructions on *how* to perform a particular task. However, we do not know how to mechanically perform more challenging tasks like translation. The field of artificial intelligence raises the level of abstraction so that we simply specify *what* the task is, and let the machine to figure out *how* to do it.

As task complexity increases, even specifying the task becomes difficult. Several criteria that we might have thought were part of a specification of fairness turn out to be provably impossible to simultaneously satisfy (Kleinberg et al., 2016; Chouldechova, 2017; Corbett-Davies et al., 2017). Reinforcement learning agents often "game" their reward function by finding solutions that technically achieve high reward without doing what the designer intended (Lehman et al., 2018; Krakovna, 2018; Clark & Amodei, 2016). In complex environments, we need to specify what *not* to change (McCarthy & Hayes, 1981); failure to do so can lead to negative side effects (Amodei et al., 2016). Powerful agents with poor specifications may pursue *instrumental subgoals* (Bostrom, 2014; Omohundro, 2008) such as resisting shutdown and accumulating resources and power (Turner, 2019).

A natural solution is to once again raise the level of abstraction, and create an agent that is uncertain about the objective and infers it from human feedback, rather than directly specifying some particular task(s). Rather than using the current model of intelligent agents optimizing for *their* objectives, we would now have beneficial agents optimizing for *our* objectives (Russell, 2019).

*Reward learning* (Leike et al., 2018; Jeon et al., 2020; Christiano et al., 2017; Ziebart et al., 2010) attempts to instantiate this by learning a reward model from human feedback, and then using a control algorithm to optimize the learned reward. Crucially, the control algorithm does not reason about the effects of the chosen actions on the reward learning process, which is external to the environment.

In contrast, in the *assistance* paradigm (Hadfield-Menell et al., 2016; Fern et al., 2014), the human $H$ is modeled as part of the environment and as having some latent goal that the agent $R$ (for robot) does not know. $R$'s goal is to maximize this (unknown) human goal. In this formulation, $R$ must balance between actions that help learn about the unknown goal, and control actions that lead to high reward.

Our key insight is that *by integrating reward learning and control modules, assistive agents can take into account the reward learning process when selecting actions*. This gives assistive agents a significant advantage over reward learning agents, which cannot perform similar reasoning.

Figure 1: $R$ must cook a pie for $H$, by placing flour on the plate to make the pie dough, filling it with either **A**pple, **B**lueberry, or **C**herry filling, and finally baking it. However, $R$ does not know which filling $H$ prefers, and $H$ is not available for questions since she is doing something else. What should $R$ do in this situation? On the right, we show what qualitative reasoning we might want $R$ to use to handle the situation.

The goal of this paper is to clarify and illustrate this advantage. We first precisely characterize the differences between reward learning and assistance, by showing that *two phase, communicative assistance* is equivalent to reward learning (Section 3). We then give qualitative examples of desirable behaviors that can only be expressed once these restrictions are lifted, and thus are only exhibited by assistive agents (Section 4).

Consider for example the kitchen environment illustrated in Figure 1, in which $R$ must bake a pie for $H$. $R$ is uncertain about which type of pie $H$ prefers to have, and currently $H$ is at work and cannot answer $R$'s questions. An assistive $R$ can make the pie crust, but wait to ask $H$ about her preferences over the filling (Section 4.1). $R$ may never clarify all of $H$'s preferences: for example, $R$ only needs to know how to dispose of food if it turns out that the ingredients have gone bad (Section 4.2). If $H$ will help with making the pie, $R$ can allow $H$ to disambiguate her desired pie by watching what filling she chooses (Section 4.3). Vanilla reward learning agents do not show these behaviors.

We do not mean to suggest that all work on reward learning should cease and only research on assistive agents should be pursued. Amongst other limitations, assistive agents are very computationally complex. Our goal is simply to clarify what qualitative benefits an assistive formulation could theoretically provide. Further research is needed to develop efficient algorithms that can capture these benefits. Such algorithms may look like algorithms designed to solve assistance problems as we have formalized them here, but they may also look like modified variants of reward learning, where the modifications are designed to provide the qualitative benefits we identify.

## 2 Background and Related Work

We introduce the key ideas behind reward learning and assistance. $X^*$ denotes a sequence of $X$. We use parametric specifications for ease of exposition, but our results apply more generally.

### 2.1 POMDPs

A **partially observable Markov decision process (POMDP)** $\mathcal{M} = \langle S, A, \Omega, O, T, r, P_0, \gamma \rangle$ consists of a finite state space $S$, a finite action space $A$, a finite observation space $\Omega$, an observation function $O : S \to \Delta(\Omega)$ (where $\Delta(X)$ is the set of probability distributions over $X$), a transition function $T : S \times A \to \Delta(S)$, a reward function $r : S \times A \times S \to \mathbb{R}$, an initial state distribution $P_0 : \Delta(S)$, and a discount rate $\gamma \in (0, 1)$. We will write $o_t$ to signify the $t$th observation $O(s_t)$. A *solution* to the POMDP is given by a policy $\pi : (O \times A)^* \times O \to \Delta(A)$ that maximizes the expected sum of rewards $ER(\pi) = \mathbb{E}_{s_0 \sim P_0, a_t \sim \pi(\cdot|o_{0:t}, a_{0:t-1}), s_{t+1} \sim T(\cdot|s_t, a_t)} \left[ \sum_{t=0}^{\infty} \gamma^t r(s_t, a_t, s_{t+1}) \right]$.

### 2.2 Reward learning

We consider two variants of reward learning: *non-active* reward learning, in which $R$ must infer the reward by observing $H$'s behavior, and *active* reward learning, in which $R$ may choose particular questions to ask $H$ in order to get particular feedback.

A **non-active reward learning problem** $\mathcal{P} = \langle \mathcal{M} \backslash r, C, \langle \Theta, r_\theta, P_\Theta \rangle, \pi^H, k \rangle$ contains a POMDP without reward $\mathcal{M} \backslash r = \langle S, A^R, \Omega^R, O^R, T, P_0, \gamma \rangle$, and instead $R$ has access to a parameterized

reward space $\langle \Theta, r_\theta, P_\Theta \rangle$. $R$ is able to learn about $\theta^*$ by observing $H$ make $k$ different choices $c$, each chosen from a set of potential choices $C$. In order for $R$ to learn from the human's choices, it also assumes access to the human decision function $\pi^H(c \mid \theta)$ that determines how the human makes choices for different possible reward functions $r_\theta$. Common decision functions include perfect optimality (Ng & Russell, 2000) and Boltzmann rationality (Ziebart et al., 2010). There are many types of choices (Jeon et al., 2020), including demonstrations (Argall et al., 2009; Ng & Russell, 2000; Ziebart et al., 2010; Fu et al., 2017; Gao et al., 2012), comparisons (Zhang et al., 2017; Wirth et al., 2017; Christiano et al., 2017; Sadigh et al., 2017), corrections (Bajcsy et al., 2017), the state of the world (Shah et al., 2019), proxy rewards (Hadfield-Menell et al., 2017b), natural language (Fu et al., 2019), etc.

A policy decision function $f(c_{0:k-1})$ produces a policy $\pi^R$ after observing $H$'s choices. A *solution* is a policy decision function $f$ that maximizes expected reward $\mathbb{E}_{\theta \sim P_\Theta, c_{0:k-1} \sim \pi^H} \left[ ER(f(c_{0:k-1})) \right]$. Since $H$'s choices $c_{0:k-1}$ do not affect the state of the environment that $R$ is acting in, this is equivalent to choosing $\pi^R$ that maximizes expected reward given the posterior over reward functions, that is $\mathbb{E}_{\theta \sim P(\theta | c_{0:k-1})} \left[ ER(\pi^R) \right]$.

An **active reward learning problem** $\mathcal{P} = \langle \mathcal{M} \backslash r, Q, C, \langle \Theta, r_\theta, P_\Theta \rangle, \pi^H, k \rangle$ adds the ability for $R$ to ask $H$ particular questions $q \in Q$ in order to get more targeted feedback about $\theta$. The human decision function $\pi^H(c \mid q, \theta)$ now depends on the question asked. A *solution* consists of a question policy $\pi_Q^R(q_i \mid q_{0:i-1}, c_{0:i-1})$ and a policy decision function $f(q_{0:k-1}, c_{0:k-1})$ that maximize expected reward $\mathbb{E}_{\theta \sim P_\Theta, q_{0:k-1} \sim \pi_Q^R, c_{0:k-1} \sim \pi^H} \left[ ER(f(q_{0:k-1}, c_{0:k-1})) \right]$.

A typical algorithm (Eric et al., 2008; Daniel et al., 2014; Maystre & Grossglauser, 2017; Christiano et al., 2017; Sadigh et al., 2017; Zhang et al., 2017; Wilde et al., 2020) will compute and ask $q \in Q$ that maximizes an *active learning criterion* such as information gain (Bıyık et al., 2019) or volume removal (Sadigh et al., 2017). Best results are achieved by selecting questions with the highest value of information (Cohn, Robert W, 2016; Zhang et al., 2017; Mindermann et al., 2018; Wilde et al., 2020), but these are usually much more computationally expensive. $R$ then finds a policy that maximizes expected reward under the inferred distribution over $\theta$, in order to approximately solve the original POMDP.

Note that a non-active reward learning problem is equivalent to an active reward learning problem with only one question, since having just a single question means that $R$ has no choice in what feedback to get (see Appendix A.1 for proofs).

## 2.3 ASSISTANCE

The key idea of assistance is that helpful behaviors like reward learning are incentivized when $R$ does not know the true reward $r$ and can only learn about it by observing human behavior. So, we model the human $H$ as part of the environment, leading to a two-agent POMDP, and assume there is some true reward $r$ that only $H$ has access to, while the robot $R$ only has access to a model relating $r$ to $H$'s behavior. Intuitively, as $R$ acts in the environment, it will also observe $H$'s behavior, which it can use to make inferences about the true reward.

Following Hadfield-Menell et al. (2016)[1], we define an **assistance game** $\mathcal{M}$ as a tuple

$$\mathcal{M} = \langle S, \{A^H, A^R\}, \{\Omega^H, \Omega^R\}, \{O^H, O^R\}, T, P_S, \gamma, \langle \Theta, r_\theta, P_\Theta \rangle \rangle.$$

Here $S$ is a finite set of states, $A^H$ a finite set of actions for $H$, $\Omega^H$ a finite set of observations for $H$, and $O^H : S \to \Delta(\Omega^H)$ an observation function for $H$ (respectively $A^R, \Omega^R, O^R$ for $R$). The transition function $T : S \times A^H \times A^R \to \Delta(S)$ gives the probability over next states given the current state and both actions. The initial state is sampled from $P_S \in \Delta(S)$. $\Theta$ is a set of possible reward function parameters $\theta$ which parameterize a class of reward functions $r_\theta : S \times A^H \times A^R \times S \to \mathbb{R}$, and $P_\theta$ is the distribution from which $\theta$ is sampled. $\gamma \in (0, 1)$ is a discount factor.

As with POMDPs, policies can depend on history. Both $H$ and $R$ are able to observe each other's actions, and on a given timestep, $R$ acts before $H$. We use $\tau_t^R : (\Omega^R \times A^H \times A^R)^t$ to denote

---

[1]Relative to Hadfield-Menell et al. (2016), our definition allows for partial observability and requires that the initial distribution over $S$ and $\Theta$ be independent. We also have $H$ choose her action sequentially after $R$, rather than simultaneously with $R$, in order to better parallel the reward learning setting.

$R$'s observations until time $t$, and $\tau_t^H$ for $H$'s observations; thus $R$'s policy can be written as $\pi^R(a^R \mid o_t^R, \tau_{t-1}^R)$, while $H$'s can be written as $\pi^H(a^H \mid o_t^H, a_t^R \tau_{t-1}^H, \theta)$. Note that unlike $H$, $R$ *does not* observe the reward parameter $\theta$, and must infer $\theta$ much like it does the hidden state.

A **fully observable assistance game** is one in which both $H$ and $R$ can observe the full state. In such cases, we omit $\Omega^H, \Omega^R, O^H$ and $O^R$.

Since we have not yet specified how $H$ behaves, it is not clear what the agent should optimize for. Should it be playing a Nash strategy or optimal strategy pair of the game, and if so, which one? Should it use a non-equilibrium policy, since humans likely do not use equilibrium strategies? This is a key hyperparameter in assistance games, as it determines the *communication protocol* for $H$ and $R$. For maximum generality, we can equip the assistance game with a *policy-conditioned belief* $B : \Pi^R \to \Delta(\Pi^H)$ over $\pi^H$, which specifies how the human responds to the agent's choice of policy (Halpern & Pass, 2018). The agent's goal is to maximize expected reward given this belief.

Prior work on assistance games (Hadfield-Menell et al., 2016; Malik et al., 2018; Woodward et al., 2019) focuses on finding optimal strategy pairs. This corresponds to a belief that $H$ will know and perfectly respond to $R$'s policy (see Appendix A.3). However, our goal is to compare assistance to reward learning. Typical reward learning algorithms assume access to a model of human decision-making: for example, $H$ might be modeled as *optimal* (Ng & Russell, 2000) or *Boltzmann-rational* (Ziebart et al., 2010). As a result, we also assume that we have access to a model of human decision-making $\pi^H$. Note that $\pi^H$ *depends on* $\theta$: we are effectively assuming that we know how $H$ *chooses how to behave* given a particular reward $r_\theta$. This assumption corresponds to the policy-conditioned belief $B(\pi_R)(\tilde{\pi}^H) = \mathbb{1}[\tilde{\pi}^H = \pi^H]$. We define an **assistance problem** $\mathcal{P}$ as a pair $\langle \mathcal{M}, \pi^H \rangle$ where $\pi^H$ is a human policy for the assistance game $\mathcal{M}$.

Given an assistance problem, a robot policy $\pi_R$ induces a probability distribution over trajectories: $\tau \sim \langle s_0, \theta, \pi^H, \pi^R \rangle, \tau \in [S \times A^H \times A^R]^*$. We denote the support of this distribution by $\text{Traj}(\pi_R)$. The *expected reward* of a robot policy for $\langle \mathcal{M}, \pi^H \rangle$ is given by

$$ER(\pi^R) = \mathop{\mathbb{E}}_{s_0 \sim P_S, \theta \sim P_\theta, \tau \sim \langle s_0, \theta, \pi^H, \pi^R \rangle} \left[ \sum_{t=0}^{\infty} \gamma^t r_\theta(s_t, a_t^H, a_t^R, s_{t+1}) \right].$$

A *solution* of $\langle \mathcal{M}, \pi^H \rangle$ is a robot policy that maximizes expected reward: $\pi^R = \operatorname*{argmax}_{\tilde{\pi}^R} ER(\tilde{\pi}^R)$.

### 2.3.1 SOLVING ASSISTANCE PROBLEMS

Once the $\pi^H$ is given, $H$ can be thought of as an aspect of the environment, and $\theta$ can be thought of as a particularly useful piece of information for estimating how good actions are. This suggests that we can reduce the assistance problem to an equivalent POMDP. Following Desai (2017), the key idea is to embed $\pi^H$ in the transition function $T$ and embed $\theta$ in the state.

In theory, to embed potentially non-Markovian $\pi^H$ in $T$, we need to embed the entire history of the trajectory in the state, but this leads to extremely large POMDPs. In our experiments, we only consider Markovian human policies, for which we do not need to embed the full history, keeping the state space manageable. Thus, the policy can be written as $\pi^H(a^H \mid o^H, a^R, \theta)$. To ensure that $R$ must infer $\theta$ from human behavior, as in the original assistance game, the observation function does *not* reveal $\theta$, but *does* reveal the previous human action $a^H$.

**Proposition 1.** *Every assistance problem $\langle \mathcal{M}, \pi^H \rangle$ can be reduced to an equivalent POMDP $\mathcal{M}'$.*

The full reduction and proof of equivalence is given in Appendix A.2.

When $\mathcal{M}$ is fully observable, in the reduced POMDP $\theta$ is the only part of the state not directly observable to the robot, making it an instance of a *hidden-goal MDP* (Fern et al., 2014). For computational tractability, much of the work on hidden goals (Javdani et al., 2015; Fern et al., 2014) selects actions *assuming that all goal ambiguity is resolved in one step*. This effectively separates reward learning and control in the same way as typical reward learning algorithms, thus negating many of the benefits we highlight in this work. Intention-aware motion planning (Bandyopadhyay et al., 2013) also embeds the human goal in the state in order to avoid collisions with humans during motion planning, but does not consider applications for assistance.

Macindoe et al. (2012) uses the formulation of a POMDP with a hidden goal to produce an assistive agent in a cops and robbers gridworld environment. Nikolaidis et al. (2015) assumes a dataset of joint human-robot demonstrations, which they leverage to learn "types" of humans that can then be inferred online using a POMDP framework. This is similar to solving an assistance problem, where we think of the different values of $\theta$ as different "types" of humans. Chen et al. (2018) uses an assistance-style framework in which the unknown parameter is the human's trust in the robot (rather than the reward $\theta$). Woodward et al. (2019) uses deep reinforcement learning to solve an assistance game in which the team must collect either plums or lemons. To our knowledge, these are the only prior works that use an assistive formulation in a way that does not ignore the information-gathering aspect of actions. While these works typically focus on algorithms to solve assistance games, we instead focus on the qualitative benefits of using an assistance formulation.

Since we can reduce an assistance problem to a regular POMDP, we can use any POMDP solver to find the optimal $\pi^R$. In our examples for this paper, we use an exact solver when feasible, and point-based value iteration (PBVI) (Pineau et al., 2003) or deep reinforcement learning (DRL) when not. When using DRL, we require recurrent models, since the optimal policy can depend on history.

A common confusion is to ask how DRL can be used, given that it requires a reward signal, but by assumption $R$ does not know the reward function. This stems from a misunderstanding of what it means for $R$ "not to know" the reward function. When DRL is run, at the beginning of each episode, a specific value of $\theta$ is sampled as part of the initial state. The *learned policy* $\pi^R$ is not provided with $\theta$: it can only see its observations $o^R$ and human actions $a^H$, and so it is accurate to say that $\pi^R$ "does not know" the reward function. However, the reward is calculated by the DRL algorithm, not by $\pi^R$, and the algorithm can and does use the sampled value of $\theta$ for this computation. $\pi^R$ can then implicitly learn the correlation between the actions $a^H$ chosen by $\pi^H$, and the high reward values that the DRL algorithm computes; this can be often be thought of as an implicit estimation of $\theta$ in order to choose the right actions.

## 3 Reward learning as two-phase communicative assistance

There are two key differences between reward learning and assistance. First, reward learning algorithms split reward learning and control into two separate phases, while assistance merges them into a single phase. Second, in reward learning, the human's only role is to communicate reward information to the robot, while in assistance the human can help with the task. These two properties exactly characterize the difference between the two: reward learning problems and communicative assistance problems with two phases can be reduced to each other, in a very natural way.

A **communicative assistance problem** is one in which the transition function $T$ and the reward function $r_\theta$ are independent of the choice of human action $a^H$, and the human policy $\pi^H(\cdot \mid o^H, a^R, \theta)$ is independent of the observation $o^H$. Thus, in a communicative assistance problem, $H$'s actions only serve to respond to $R$, and have no effects on the state or the reward (other than by influencing $R$). Such problems can be cast as instances of HOP-POMDPs (Rosenthal & Veloso, 2011).

For the notion of two phases, we will also need to classify robot actions as communicative or not. We will assume that there is some distinguished action $a^R_{noop}$ that "does nothing". Then, a robot action $\hat{a}^R$ is **communicative** if for any $s, a^H, s'$ we have $T(s' \mid s, a^H, \hat{a}^R) = T(s' \mid s, a^H, a^R_{noop})$ and $R(s, a^H, \hat{a}^R, s') = R(s, a^H, a^R_{noop}, s')$. A robot action is **physical** if it is not communicative.

Now consider a communicative assistance problem $\langle \mathcal{M}, \pi^H \rangle$ with noop action $a^R_{noop}$ and let the optimal robot policy be $\pi^{R*}$. Intuitively, we would like to say that there is an initial communication phase in which the only thing that happens is that $H$ responds to questions from $R$, and then a second action phase in which $H$ does nothing and $R$ acts. Formally, the assistance problem is **two phase with actions at $t_{act}$** if it satisfies the following property:

$$\exists a^H_{noop} \in A^H, \ \forall \tau \in \text{Traj}(\pi^{R*}), \ \left[ \forall t < t_{act} : a^R_t \text{ is communicative } \wedge \forall t \geq t_{act} : \ a^H_t = a^H_{noop} \right].$$

Thus, in a two phase assistance problem, every trajectory from an optimal policy can be split into a "communication" phase where $R$ cannot act and an "action" phase where $H$ cannot communicate.

**Reducing reward learning to assistance.** We can convert an active reward learning problem to a two-phase communicative assistance problem in an intuitive way: we add $Q$ to the set of robot actions,

make $C$ the set of human actions, add a timestep counter to the state, and construct the reward such that an optimal policy must switch between the two phases after $k$ questions. A non-active reward learning problem can first be converted to an active reward learning problem.

**Proposition 2.** *Every active reward learning problem $\langle \mathcal{M}, Q, C, \langle \Theta, r_\theta, P_\Theta \rangle, \pi^H, k \rangle$ can be reduced to an equivalent two phase communicative assistance problem $\langle \mathcal{M}', \pi^{H'} \rangle$.*

**Corollary 3.** *Every non-active reward learning problem $\langle \mathcal{M}, C, \langle \Theta, r_\theta, P_\Theta \rangle, \pi^H, k \rangle$ can be reduced to an equivalent two phase communicative assistance problem $\langle \mathcal{M}', \pi^{H'} \rangle$.*

**Reducing assistance to reward learning.** The reduction from a two-phase communicative assistance problem to an active reward learning problem is similarly straightforward: we interpret $R$'s communicative actions as questions and $H$'s actions as answers. There is once again a simple generalization to non-active reward learning.

**Proposition 4.** *Every two-phase communicative assistance problem $\langle \mathcal{M}, \pi^H, a_{noop}^R \rangle$ can be reduced to an equivalent active reward learning problem.*

**Corollary 5.** *If a two-phase communicative assistance problem $\langle \mathcal{M}, \pi^H \rangle$ has only one communicative robot action, it can be reduced to an equivalent non-active reward learning problem.*

## 4 QUALITATIVE IMPROVEMENTS FOR GENERAL ASSISTANCE

We have seen that reward learning is equivalent to two-phase communicative assistance problems, where inferring the reward distribution can be separated from control using the reward distribution. However, for general assistance games, it is necessary to merge estimation and control, leading to several new qualitative behaviors. When the two phase restriction is lifted, we observe *relevance aware active learning* and *plans conditional on future feedback*. When the communicative restriction is lifted, we observe *learning from physical actions*.

We demonstrate these qualitative behaviors in simple environments using point-based value iteration (PBVI) or deep reinforcement learning (DRL). We describe the qualitative results here, deferring detailed explanations of environments and results to Appendix C. For communicative assistance problems, we also consider two baselines:

1. **Active reward learning.** This is the reward learning paradigm discussed so far.
2. **Interactive reward learning.** This is a variant of reward learning that aims to recover some of the benefits of interactivity, by alternating reward learning and acting phases. During an action phase, $R$ chooses actions that maximize expected reward *under its current belief over $\theta$* (without "knowing" that its belief may change), while during a reward learning phase, $R$ chooses questions that maximizes information gain.

### 4.1 PLANS CONDITIONAL ON FUTURE FEEDBACK

Here, we show how an assistive agent can make plans that depend on obtaining information about $\theta$ in the future. The agent can first take some "preparatory" actions that whose results can be used later once the agent has clarified details about $\theta$. A reward learning agent would not be able to do this, as it would require three phases (acting, then learning, then acting again).

We illustrate this with our original kitchen environment (Figure 1), in which $R$ must bake a pie for $H$, but doesn't know what type of pie $H$ would like: **A**pple, **B**lueberry, or **C**herry. Each type has a weight specifying the reward for that pie. Assuming people tend to like apple pie the most and cherry pie the least, we have $\theta_A \sim \text{Uniform}[2, 4]$, $\theta_B \sim \text{Uniform}[1, 3]$, and $\theta_C \sim \text{Uniform}[0, 2]$. We define the questions $Q = \{q_A, q_B, q_C\}$, where $q_X$ means "What is the value of $\theta_X$?", and thus, the answer set is $C = \mathbb{R}$.

$R$ can select ingredients to assemble the pie. Eventually, $R$ must use "bake", which bakes the selected ingredients into a finished pie, resulting in reward that depends on what type of pie has been created. $H$ initially starts outside the room, but will return at some prespecified time. $r_\theta$ assigns a cost of asking a question of $0.1$ if $H$ is inside the room, and $3$ otherwise. The horizon is 6 timesteps.

**Assistance.** Notice that, *regardless of $H$'s preferences*, $R$ will need to use flour to make pie dough. So, $R$ always makes the pie dough first, before querying $H$ about her preferences. Whether $R$ then

queries $H$ about her preferences depends on how late $H$ returns. If $H$ arrives home before timestep 5, $R$ will query her about her preferences and then make the appropriate pie as expected. However, if $H$ will arrive later, then there will not be enough time to query her for her preferences and bake a pie. Instead, $R$ bakes an apple pie, since its prior suggests that's what $H$ wants.

This behavior, where $R$ takes actions (making dough) that are robustly good but waits on actions (adding the filling) whose reward will be clarified in the future, is very related to *conservative agency* (Turner et al., 2020), a connection explored in more depth in Appendix D.

**Reward learning.** The assistance solution requires $R$ to act (to make dough), then to learn preferences, and then to act again (to make pie). A reward learning agent can only have two phases, and so we see one of two suboptimal behaviors. First, $R$ could stay in the learning phase until $H$ returns home, then ask which pie she prefers, and then make the pie from scratch. Second, $R$ could make an apple pie without asking $H$ her preferences. (In this case there would be no learning phase.) Which of these happens depends on the particular method and hyperparameters used.

**Interactive reward learning.** Adding interactivity is not sufficient to get the correct behavior. Suppose we start with an action phase. The highest reward plan under $R$'s current belief over $\theta$ is to bake an apple pie, so that's what it will do, as long as the phase lasts long enough. Conversely, suppose we start with a learning phase. In this case, $R$ does nothing until $H$ returns, and then asks about her preferences. Once we switch to an action phase, it bakes the appropriate pie from scratch.

## 4.2 RELEVANCE AWARE ACTIVE LEARNING

Once we relax the two-phase restriction, $R$ starts to further optimize *whether* and *when* it asks questions. In particular, since $R$ may be uncertain about whether a question's answer will even be necessary, $R$ will only ask questions once they become *immediately relevant* to the task at hand. In contrast, a reward learning agent would have to decide at the beginning of the episode (during the learning phase) whether or not to ask these questions, and so cannot evaluate how relevant they are.

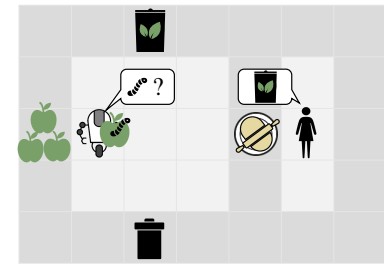

Figure 2: The wormy-apples kitchen environment. $H$ wants an apple, but $R$ might discover worms in the apple, and have to dispose of it in either of the trash or compost bins.

Consider for example a modification to the kitchen environment: $R$ knows that $H$ wants an apple pie, but when $R$ picks up some apples, there is a 20% chance that it finds worms in some of the apples. $R$ is unsure whether $H$ wants her compost bin to have worms, and so does not know whether to dispose of the bad apples in the trash or compost bin. Since this situation is relatively unlikely, ideally $R$ would only clarify $H$'s preferences when the situation arises.

**Assistance.** An assistive $R$ only asks about wormy apples when it needs to dispose of one. $R$ always starts by picking up apples. If the apple does not have worms, $R$ immediately uses the apples to bake the pie. If some apples have worms and the cost of asking a question is sufficiently low, $R$ elicits $H$'s preferences and disposes of the apples appropriately. It then bakes the pie with the remaining apples.

This behavior, in which questions are asked only if they are useful for constraining future behavior, has been shown previously using probabilistic recipe trees (PRTs) Kamar et al. (2009), but to our knowledge has not been shown with optimization-based approaches.

**Reward learning.** A reward learning policy must have only two phases and so would show one of two undesirable behaviors: either it would always ask $H$ where to dispose of wormy apples, or it never asks and instead guesses when it does encounter wormy apples.

**Interactive reward learning.** This has the same problem as in the previous section. If we start in the action phase and $R$ picks up wormy apples, it will dispose of them in an arbitrary bin without asking $H$ about her preferences, because it doesn't "know" that it will get the opportunity to do so. Alternatively, if we start with a learning phase, $R$ will ask $H$ where to dispose of wormy apples, even if $R$ would never pick up any wormy apples.

Note that more complex settings can have *many* more questions. Should $R$ ask whether $H$ would prefer to use seedless apples, should scientists ever invent them? Perhaps $R$ should ask $H$ how her pie preferences vary based on her emotional state? Asking about all possible situations is not scalable.

### 4.3 LEARNING FROM PHYSICAL ACTIONS

So far we have considered *communicative* assistance problems, in which $H$ only provides feedback rather than acting to maximize reward herself. Allowing $H$ to have physical actions enables a greater variety of potential behaviors. Most clearly, when $R$ knows the reward (that is, $P_\Theta$ puts support over a single $\theta$), assistance games become equivalent to human-AI collaboration (Nikolaidis & Shah, 2013; Carroll et al., 2019; Dimitrakakis et al., 2017).

With uncertain rewards, we can see further interesting qualitative behaviors: $R$ can learn just by observing how $H$ acts in an environment, and then work with $H$ to maximize reward, *all within a single episode*, as in shared autonomy with intent inference (Javdani et al., 2015; Brooks & Szafir, 2019) and other works that interpret human actions as communicative Whitney et al. (2017). This can significantly reduce the burden on $H$ in providing reward information to $R$ (or equivalently, reduce the cost incurred by $R$ in asking questions to $H$). Some work has shown that in such situations, humans tend to be *pedagogic*: they knowingly take individually suboptimal actions, in order to more effectively convey the goal to the agent (Ho et al., 2016; Hadfield-Menell et al., 2016). An assistive $R$ who knows this can quickly learn what $H$ wants, and help her accomplish her goals.

We illustrate this with a variant of our kitchen environment, shown in Figure 3. There are no longer questions and answers. Both $H$ and $R$ can move to an adjacent free space, and pick up and place the various objects. Only $R$ may bake the dessert. $R$ is uncertain whether $H$ prefers cake or cherry pie.

For both recipes, it is individually more efficient for $H$ to pick up the dough first. However, we assume $H$ is pedagogic and wants to quickly show $R$ which recipe she wants. So, if she wants cake, she will pick up the chocolate first to signal to $R$ that cake is the preferred dessert.

It is not clear how exactly to think about this from a reward learning perspective: there aren't any communicative human actions since every action alters the state of the environment. In addition, there is no clear way to separate out a given trajectory into two phases. This situation cannot be easily coerced into the reward learning paradigm.

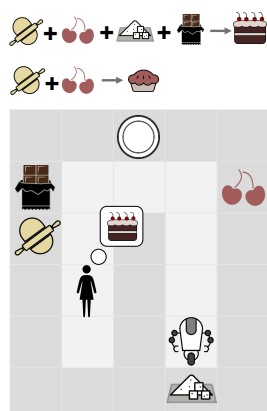

Figure 3: The cake-or-pie variant of the kitchen environment. $H$ is equally likely to prefer cake or pie. Communication must take place through physical actions alone.

In contrast, an assistive $R$ can handle this situation perfectly. It initially waits to see which ingredient $H$ picks up first, and then quickly helps $H$ by putting in the ingredients from its side of the environment and baking the dessert. It learns implicitly to make the cake when $H$ picks up chocolate, and to make the pie when $H$ picks up dough. This is equivalent to pragmatic reasoning (Goodman & Frank, 2016): "$H$ would have picked up the chocolate if she wanted cake, so the fact that she picked up the dough implies that she wants cherry pie". However, we emphasize that $R$ is not explicitly programmed to reason in this manner, and is learned using deep reinforcement learning (Appendix C.3).

Note that $R$ is not limited to learning from $H$'s physical actions: $R$ can also use its own physical actions to "query" the human for information (Woodward et al., 2019; Sadigh et al., 2016).

## 5 LIMITATIONS AND FUTURE WORK

**Computational complexity.** The major limitation of assistance compared to reward learning is that assistance problems are significantly more computationally complex, since we treat the unknown reward $\theta$ as the hidden state of a POMDP. We are hopeful that this can be solved through the application of deep reinforcement learning. An assistance problem is just like any other POMDP,

except that there is one additional unobserved state variable $\theta$ and one additional observation $a^H$. This should not be a huge burden, since deep reinforcement learning has been demonstrated to scale to huge observation and action spaces (OpenAI, 2018; Vinyals et al., 2019). Another avenue for future work is to modify active reward learning algorithms in order to gain the benefits outlined in Section 4, while maintaining their computational efficiency.

**Increased chance of incorrect inferences.** In practice, assistive agents will extract more information from $H$ than reward learning agents, and so it is worse if $\pi^H$ is misspecified. We don't see this as a major limitation: to the extent this is a major worry, we can design $\pi^H$ so that the robot only makes inferences about human behavior in specific situations. For example, by having $\pi^H$ be independent of $\theta$ in a given state $s$, we ensure that the robot does not make any inferences about $\theta$ in that state.

**Environment design.** We have shown that by having a hidden human goal, we can design environments in which optimal agent behavior is significantly more "helpful". One important direction for future work is to design larger, more realistic environments, in order to spur research into how best to solve such environments. We would be particularly excited to see a suite of assistance problems become a standard benchmark by which deep reinforcement learning algorithms are assessed.

### 5.1 LIMITATIONS OF ASSISTANCE AND REWARD LEARNING

While we believe that the assistance framework makes meaningful conceptual progress over reward learning, a number of challenges for reward learning remain unaddressed by assistance:

**Human modeling.** A major motivation for both paradigms is that reward specification is very difficult. However, now we need to specify a prior over reward functions, and the human model $\pi^H$. Consequently, misspecification can still lead to bad results (Armstrong et al., 2020; Carey, 2018). While it should certainly be easier to specify a prior over $\theta$ with a "grain of truth" on the true reward $\theta^*$ than to specify $\theta^*$ directly, it is less clear that we can specify $\pi^H$ well.

One possibility is to add uncertainty over the human policy $\pi^H$. However, this can only go so far: information about $\theta$ must come from *somewhere*. If $R$ is sufficiently uncertain about $\theta$ and $\pi^H$, then it cannot learn about the reward (Armstrong & Mindermann, 2018). Thus, for good performance we need to model $\pi^H$. While imitation learning can lead to good results (Carroll et al., 2019), the best results will likely require insights from a broad range of fields that study human behavior.

**Assumption that $H$ knows $\theta$.** Both assistance games and reward learning makes the assumption that $H$ knows her reward exactly, but in practice, human preferences change over time (Allais, 1979; Cyert & DeGroot, 1975; Shogren et al., 2000). We could model this as the human changing their subgoals (Michini & How, 2012; Park et al., 2020), adapting to the robot (Nikolaidis et al., 2017) or learning from experience (Chan et al., 2019).

**Dependence on uncertainty.** All of the behaviors of Section 4, as well as previously explored benefits such as off switch corrigibility (Hadfield-Menell et al., 2017a), depend on $R$ expecting to gain information about $\theta$. However, $R$ will eventually exhaust the available information about $\theta$. If everything is perfectly specified, this is not a problem: $R$ will have converged to the true $\theta^*$. However, in the case of misspecification, after convergence $R$ is effectively certain in an incorrect $\theta$, which has many troubling problems that we sought to avoid in the first place (Yudkowsky, year unknown).

## 6 CONCLUSION

While much recent work has focused on how we can build agents that learn what they should do from human feedback, there is not yet a consensus on how such agents should be built. In this paper, we contrasted the paradigms of *reward learning* and *assistance*. We showed that reward learning problems are equivalent to a special type of assistance problem, in which the human may only provide feedback at the beginning of the episode, and the agent may only act in the environment after the human has finished providing feedback. By relaxing these restrictions, we enable the agent to reason about how its actions in the environment can influence the process by which it solicits and learns from human feedback. This allows the agent to (1) choose questions based on their relevance, (2) create plans whose success depends on future feedback, and (3) learn from physical human actions in addition to communicative feedback.

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

## A    REWARD LEARNING AND ASSISTANCE FORMALISMS

### A.1    RELATION BETWEEN NON-ACTIVE AND ACTIVE REWARD LEARNING

The key difference between non-active and active reward learning is that in the latter $R$ may ask $H$ questions in order to get more targeted feedback. This matters as long as there is more than one question: with only one question, since there is no choice for $R$ to make, $R$ cannot have any influence on the feedback that $H$ provides. As a result, non-active reward learning is equivalent to active reward learning with a single question.

**Proposition 6.** *Every non-active reward learning problem $\langle \mathcal{M} \backslash r, C, \langle \Theta, r_\theta, P_\Theta \rangle, \pi^H, k \rangle$ can be reduced to an active reward learning problem.*

*Proof.* We construct the active reward learning problem as $\langle \mathcal{M} \backslash r, Q', C, \langle \Theta, r_\theta, P_\Theta \rangle, \pi^{H'}, k \rangle$, where $Q' \triangleq \{q_\phi\}$ where $q_\phi$ is some dummy question, and $\pi^{H'}(c \mid q, \theta) \triangleq \pi^H(c \mid \theta)$.

Suppose the solution to the new problem is $\langle \pi_Q^{R'}, f' \rangle$. Since $f'$ is a solution, we have:

$$
\begin{aligned}
f' &= \operatorname*{argmax}_{\hat{f}} \quad \mathbb{E}_{\theta \sim P_\Theta, q_{0:k-1} \sim \pi_Q^{R'}, c_{0:k-1} \sim \pi^{H'}(\cdot | q_i, \theta)} \left[ ER(\hat{f}(q_{0:k-1}, c_{0:k-1})) \right] \\
&= \operatorname*{argmax}_{\hat{f}} \quad \mathbb{E}_{\theta \sim P_\Theta, q_{0:k-1} = q_\phi, c_{0:k-1} \sim \pi^{H'}(\cdot | q_\phi, \theta)} \left[ ER(\hat{f}(q_{0:k-1} = q_\phi, c_{0:k-1})) \right] \quad \text{all } q \text{ are } q_\phi \\
&= \operatorname*{argmax}_{\hat{f}} \quad \mathbb{E}_{\theta \sim P_\Theta, c_{0:k-1} \sim \pi^H(\cdot | \theta)} \left[ ER(\hat{f}(q_{0:k-1} = q_\phi, c_{0:k-1})) \right].
\end{aligned}
$$

Thus $f(c_{0:k-1}) = f'(q_{0:k-1} = q_\phi, c_{0:k-1})$ is a maximizer of $\mathbb{E}_{\theta \sim P_\Theta, c_{0:k-1} \sim \pi^H(\cdot | \theta)} \left[ ER(\hat{f}(c_{0:k-1})) \right]$, making it a solution to our original problem. $\square$

**Proposition 7.** *Every active reward learning problem $\langle \mathcal{M} \backslash r, Q, C, \langle \Theta, r_\theta, P_\Theta \rangle, \pi^H, k \rangle$ with $|Q| = 1$ can be reduced to a non-active reward learning problem.*

*Proof.* Let the sole question in $Q$ be $q_\phi$. We construct the non-active reward learning problem as $\langle \mathcal{M} \backslash r, C, \langle \Theta, r_\theta, P_\Theta \rangle, \pi^{H'}, k \rangle$, with $\pi^{H'}(c \mid \theta) = \pi^H(c \mid q_\phi, \theta)$.

Suppose the solution to the new problem is $f'$. Then we can construct a solution to the original problem as follows. First, note that $\pi_Q^R$ must be $\pi_Q^R(q_i \mid q_{0:i-1}, c_{0:i-1}) = \mathbb{1}[q_i = q_\phi]$, since there is only one possible question $q_\phi$. Then by inverting the steps in the proof of Proposition 6, we can see that $f'$ is a maximizer of $\mathbb{E}_{\theta \sim P_\Theta, q_{0:k-1} \sim \pi_Q^R, c_{0:k-1} \sim \pi^H(\cdot | q_i, \theta)} \left[ ER(\hat{f}(\cdot \mid c_{0:k-1})) \right]$. Thus, by defining $f(q_{0:k-1}, c_{0:k-1}) = f'(c_{0:k-1})$, we get a maximizer to our original problem, making $\langle \pi_Q^R, f \rangle$ a solution to the original problem. $\square$

### A.2    REDUCING ASSISTANCE PROBLEMS TO POMDPS

Suppose that we have an assistance problem $\langle \mathcal{M}, \pi^H \rangle$ with:

$$
\mathcal{M} = \langle S, \{A^H, A^R\}, \{\Omega^H, \Omega^R\}, \{O^H, O^R\}, T, P_S, \gamma, \langle \Theta, r_\theta, P_\Theta \rangle \rangle.
$$

Then, we can derive a single-player POMDP for the robot $\mathcal{M}' = \langle S', A^R, \Omega', O', T', r', P_0', \gamma \rangle$ by embedding the human reward parameter into the state. We must include the human's *previous* action $a^H$ into the state, so that the robot can observe it, and so that the reward can be computed.

To allow for arbitrary (non-Markovian) human policies $\pi^H$, we could encode the full history in the state, in order to embed $\pi^H$ into the transition function $T$. However, in our experiments we only consider human policies that are in fact Markovian. We make the same assumption here, giving a policy $\pi^H(a_t^H \mid o_t^H, a_t^R, \theta)$ that depends on the current observation and previous robot action.

The transformation $\mathcal{M} \mapsto \mathcal{M}'$ is given as follows:

$$
\begin{aligned}
S' &\triangleq S \times A^H \times \Theta & \text{State space}\\
\Omega' &\triangleq \Omega^R \times A^H & \text{Observation space}\\
O'(o' \mid s') &= O'((o^R, a_1^H) \mid (s, a_2^H, \theta)) & \text{Observation function}\\
&\triangleq \mathbb{1}[a_1^H = a_2^H] \cdot O^R(o^R \mid s)\\
T'(s_2' \mid s_1', a^R) &= T'((s_2, a_1^H, \theta_2) \mid (s_1, a_0^H, \theta_1), a^R) & \text{Transition function}\\
&\triangleq T(s_2 \mid s_1, a_1^H, a^R) \cdot \mathbb{1}[\theta_2 = \theta_1] \cdot \sum_{o^H \in \Omega^H} O^H(o^H \mid s_1) \cdot \pi^H(a_1^H \mid o^H, a^R, \theta)\\
r'(s_1', a^R, s_2') &= r'((s_1, a_0^H, \theta), a^R, (s_2, a_1^H, \theta)) & \text{Reward function}\\
&\triangleq r_\theta(s_1, a_1^H, a^R, s_2)\\
P_0'(s') &= P_0'((s, a^H, \theta)) & \text{Initial state distribution}\\
&\triangleq P_S(s) \cdot P_\Theta(\theta) \cdot \mathbb{1}[a^H = a_{init}^H] & \text{where } a_{init}^H \text{ is arbitrary}
\end{aligned}
$$

In the case where the original assistance problem is fully observable, the resulting POMDP is an instance of a Bayes-Adaptive MDP (Martin, 1967; Duff, 2002).

Any robot policy $\pi^R$ can be translated from the APOMDP $\mathcal{M}$ naturally into an identical policy on $\mathcal{M}'$. Note that in either case, policies are mappings from $(\Omega^R, A^H, A^R)^* \times \Omega^R$ to $\Delta(A^R)$.

This transformation preserves optimal agent policies:

**Proposition 8.** *A policy $\pi^R$ is a solution of $\mathcal{M}$ if and only if it is a solution of $\mathcal{M}'$.*

*Proof.* Recall that an optimal policy $\pi^*$ in the POMDP $\mathcal{M}'$ is one that maximizes the expected value:

$$
\mathrm{EV}(\pi) = \mathop{\mathbb{E}}_{s_0' \sim P_0', \tau' \sim \langle s_0', \pi \rangle} \left[ \sum_{t=0}^\infty \gamma^t r'(s_t', a_t, s_{t+1}) \right] = \mathop{\mathbb{E}}_{s_0' \sim P_0', \tau' \sim \langle s_0', \pi \rangle} \left[ \sum_{t=0}^\infty \gamma^t r_\theta(s_t, a_t^H, a_t, s_{t+1}) \right]
$$

where the trajectories $\tau'$s are sequences of state, action pairs drawn from the distribution induced by the policy, starting from state $s_0$.

Similarly, an optimal robot policy $\pi^{R*}$ in the APOMDP $\mathcal{M}$ is one that maximizes its expected reward:

$$
\mathrm{ER}(\pi^R) = \mathop{\mathbb{E}}_{s_0 \sim P_S, \theta \sim P_\Theta, \tau \sim \langle s_0, \theta, \pi^R \rangle} \left[ \sum_{t=0}^\infty \gamma^t r_\theta(s_t, a_t^H, a_t^R, s_{t+1}) \right].
$$

To show that the optimal policies coincide, suffices to show that for any $\pi$, $\mathrm{ER}(\pi)$ (in $\mathcal{M}$) is equal to $\mathrm{EV}(\pi)$ (in $\mathcal{M}'$). To do this, we will show that $\pi$ induces the "same" distributions over the trajectories. For mathematical convenience, we will abuse notation and consider trajectories of the form $\tau; \theta \in (S, A^H, A^R)^* \times \Theta$; it is easy to translate trajectories of this form to trajectories in either $\mathcal{M}'$ or $\mathcal{M}$.

We will show that the sequence $\tau; \theta$ has the same probability when the robot takes the policy $\pi$ in both $\mathcal{M}'$ and $\mathcal{M}$ by induction on the lengths of the sequence.

First, consider the case of length 1 sequences. $\tau; \theta = [(s, a^R, a^H); \theta]$. Under both $\mathcal{M}'$ and $\mathcal{M}$, $s$ and $\theta$ are drawn from $P_S$ and $P_\Theta$ respectively. Similarly, $a^R$ and $a^H$ are drawn from $\pi^R(\cdot \mid o_0^R)$ and $\pi^H(\cdot \mid o^H, a^R, \theta)$ respectively. So the distribution of length 1 sequences is the same under both $\mathcal{M}'$ and $\mathcal{M}$.

Now, consider some longer sequence $\tau; \theta = [(s_1, a_1^R, a_1^H), ...., (s_t, a_t^R, a_t^H); \theta]$. By the inductive hypothesis, the distribution of $(s_1, a_1^H, a_1^R), ...., (s_{t-1}, a_{t-1}^H, a_{t-1}^R)$ and $\theta$ are identical; it suffices to show that $(s_t, a_t^H, a_t^R)$ has the same distribution, conditioned on the other parts of $\tau; \theta$, under $\mathcal{M}'$ and under $\mathcal{M}$. Yet by construction, $s_t$ is drawn from the same distribution $T(\cdot \mid s_{t-1}, a_{t-1}^H, a_{t-1}^R)$, $a_t^H$ is drawn from the same distribution $\pi^H(\cdot \mid o_t^H, a_t^R, \theta)$, and $a_t^R$ is drawn from the same distribution $\pi^R(\cdot \mid o_t^R, \tau_{t-1}^R)$. $\square$

### A.3 Optimal strategy pairs as policy-conditioned belief

We use the term *policy-conditioned belief* to refer to a distribution over human policies which depends on the chosen robot policy. We use policy-conditioned beliefs as opposed to a simple unconditional distribution over human policies, because it allows us to model a wide range of situations, including situations with prior coordination, or where humans adapt to the robot's policy as a result of prior interactions. Moreover, this presents a unifying framework with prior work on assistance games (Hadfield-Menell et al., 2016). In fact, finding an optimal strategy pair for the assistance game can be thought of as finding the policy which is best when the human adapts optimally, as formalized below:

**Proposition 9.** *Let $\mathcal{M} = \langle S, \{A^H, A^R\}, \{\Omega^H, \Omega^R\}, \{O^H, O^R\}, T, P_S, \gamma, \langle \Theta, r_\theta, P_\Theta \rangle \rangle$ be an assistance game. Let $B(\pi^R)(\pi^H) \propto \mathbb{1}[EJR(\pi^H, \pi^R) = \max_{\tilde{\pi}^H \in \Pi^H} EJR(\tilde{\pi}^H, \pi^R)]$ be an associated policy-conditioned belief. Let $\pi^R$ be the solution to $\langle \mathcal{M}, B \rangle$. Then $\langle B(\pi^R), \pi^R \rangle$ is an optimal strategy pair.*

*Proof.* Let $\langle \overline{\pi}^H, \overline{\pi}^R \rangle$ be an arbitrary strategy pair. Then $EJR(\overline{\pi}^H, \overline{\pi}^R) \leq EJR(B(\overline{\pi}^R), \overline{\pi}^R)$ by the definition of $B$, and $EJR(B(\overline{\pi}^R), \overline{\pi}^R) \leq EJR(B(\pi^R), \pi^R)$ by the definition of $\pi^R$. Thus $EJR(\overline{\pi}^H, \overline{\pi}^R) \leq EJR(B(\pi^R), \pi^R)$. Since $\langle \overline{\pi}^H, \overline{\pi}^R \rangle$ was assumed to be arbitrary, $\langle B(\pi^R), \pi^R \rangle$ is an optimal strategy pair. $\qquad\square$

## B Equivalence of restricted assistance and existing algorithms

### B.1 Equivalence of two phase assistance and reward learning

Here we prove the results in Section 3 showing that two phase communicative assistance problems and reward learning problems are equivalent.

We first prove Proposition 4, and then use it to prove the others.

**Proposition 4.** *Every two-phase communicative assistance problem $\langle \mathcal{M}, \pi^H, a^R_{noop} \rangle$ can be reduced to an equivalent active reward learning problem.*

*Proof.* Let $\mathcal{M} = \langle S, \{A^H, A^R\}, \{\Omega^H, \Omega^R\}, \{O^H, O^R\}, T, P_S, \gamma, \langle \Theta, r_\theta, P_\Theta \rangle \rangle$ be the assistance game, and let the assistance problem's action phase start at $t_{act}$. Let $a^H_\phi \in A^H$ be some arbitrary human action and $o^H_\phi \in \Omega^H$ be some arbitrary human observation. We construct the new active reward learning problem $\langle \mathcal{M}', Q', C', \langle \Theta, r'_\theta, P_\Theta \rangle, \pi^{H'}, k' \rangle$ as follows:

$$Q' \triangleq \{a^R \in A^R : a^R \text{ is communicative}\} \qquad \text{Questions}$$
$$C' \triangleq A^H \qquad \text{Answers}$$
$$\mathcal{M}' \triangleq \langle S, A', \Omega^R, O^R, T', P'_0, \gamma \rangle \qquad \text{POMDP}$$
$$A' \triangleq A^R \backslash Q' \qquad \text{Physical actions}$$
$$T'(s' \mid s, a^R) \triangleq T(s' \mid s, a^H_\phi, a^R) \qquad \text{Transition function}$$
$$k' \triangleq t_{act} \qquad \text{Number of questions}$$
$$P'_0(s) \triangleq \sum_{s_{0:k'} \in S} P_\mathcal{M}(s_{0:k'}, s_{k'+1} = s \mid a^R_{0:k'} = a^R_{noop}, a^H_{0:k'} = a^H_\phi) \qquad \text{Initial state distribution}$$
$$r'_\theta(s, a^R, s') \triangleq r_\theta(s, a^H_\phi, a^R, s') \qquad \text{Reward function}$$
$$\pi^{H'}(c \mid q, \theta) \triangleq \pi^H(c \mid o^H_\phi, q, \theta) \qquad \text{Human decision function}$$

Note that it is fine to use $a^H_\phi$ in $T, r_\theta$ and to use $o^H_\phi$ in $\pi^H$ even though they were chosen arbitrarily, because since the assistance problem is communicative, the result does not depend on the choice.

The $P_{\mathcal{M}}$ term in the initial state distribution denotes the probability of a trajectory under $\mathcal{M}$ and can be computed as

$$P_{\mathcal{M}}(s_{0:T+1} \mid a_{0:T}^R, a_{0:T}^H) = P_S(s_0) \prod_{t=0}^{T} T(s_{t+1} \mid s_t, a_t^H, a_t^R).$$

Given some pair $\langle \pi_Q^{R'}, f' \rangle$ to the active reward learning problem, we construct a policy for the assistance problem as

$$\pi^R(a_t^R \mid o_t^R, \tau_{t-1}^R) \triangleq \begin{cases} \pi_Q^{R'}(a_t^R \mid a_{0:t-1}^R, a_{0:t-1}^H), & t < k \text{ and } a_{0:t}^R \in Q' \\ f'(a_{0:k-1}^R, a_{0:k-1}^H)(a_t^R \mid o_{k:t}^R, a_{k:t-1}^R), & t \geq k \text{ and } a_{0:k-1}^R \in Q' \text{ and } a_{k:t}^R \in A' \ . \\ 0, & \text{else} \end{cases}$$

We show that there must exist a solution to $\mathcal{P}$ that is the analogous policy to some pair. Assume towards contradiction that this is not the case, and that there is a solution $\pi^{R*}$ that is not the analogous policy to some pair. Then we have a few cases:

1. $\pi^{R*}$ assigns positive probability to $a_i^R = a \notin Q'$ for $i < k$. This contradicts the two-phase assumption.

2. $\pi^{R*}$ assigns positive probability to $a_i^R = q \in Q$ for $i \geq k$. This contradicts the two-phase assumption.

3. $\pi^{R*}(a_t^R \mid o_t^R, \tau_{t-1}^R)$ depends on the value of $o_i^R$ for some $i < k$. Since both $a_{0:k-1}^H$ and $a_{0:k-1}^R$ cannot affect the state or reward (as they are communicative), the distribution over $o_{0:k-1}^R$ is fixed and independent of $\pi^R$, and so there must be some other $\pi^R$ that is independent of $o_{0:k-1}^R$ that does at least as well. That $\pi^R$ would be the analogous policy to some pair, giving a contradiction.

Now, suppose we have some pair $\langle \pi_Q^{R'}, f' \rangle$, and let its analogous policy be $\pi^R$. Then we have:

$$\mathop{\mathbb{E}}_{\theta \sim P_\Theta, q_{0:k-1} \sim \pi_Q^{R'}, c_{0:k-1} \sim \pi^{H'}} [ER(f'(q_{0:k-1}, c_{0:k-1}))]$$

$$= \mathop{\mathbb{E}}_{\theta \sim P_\Theta} \left[ \mathop{\mathbb{E}}_{q_{0:k-1} \sim \pi^R, c_{0:k-1} \sim \pi^H} [ER(f'(q_{0:k-1}, c_{0:k-1}))] \right]$$

$$= \mathop{\mathbb{E}}_{\theta \sim P_\Theta} \left[ \mathop{\mathbb{E}}_{q_{0:k-1} \sim \pi^R, c_{0:k-1} \sim \pi^H} \left[ \mathop{\mathbb{E}}_{s_0 \sim P_0', a_t^R \sim f'(q_{0:k-1}, c_{0:k-1}), s_{t+1} \sim T'(\cdot \mid s_t, a_t^R)} \left[ \sum_{t=0}^{\infty} \gamma^t r_\theta'(s_t, a_t^R, s_{t+1}) \right] \right] \right]$$

$$= \mathop{\mathbb{E}}_{\theta \sim P_\Theta} \left[ \mathop{\mathbb{E}}_{q_{0:k-1} \sim \pi^R, c_{0:k-1} \sim \pi^H} \left[ \mathop{\mathbb{E}}_{s_k \sim P_0', a_t^R \sim \pi^R(\cdot \mid \langle c_{0:k-1}, o_{k:t} \rangle, \langle q_{0:k-1}, a_{k:t-1} \rangle), s_{t+1} \sim T'(\cdot \mid s_t, a_t^R)} \left[ \frac{1}{\gamma^k} \sum_{t=k}^{\infty} \gamma^t r_\theta'(s_t, a_t^R, s_{t+1}) \right] \right] \right]$$

$$= \mathop{\mathbb{E}}_{\theta \sim P_\Theta} \left[ \mathop{\mathbb{E}}_{q_{0:k-1} \sim \pi^R, c_{0:k-1} \sim \pi^H} \left[ \mathop{\mathbb{E}}_{s_k \sim P_0', a_t^R \sim \pi^R(\cdot \mid \langle c_{0:k-1}, o_{k:t} \rangle, \langle q_{0:k-1}, a_{k:t-1} \rangle), s_{t+1} \sim T'(\cdot \mid s_t, a_t^R)} \left[ \frac{1}{\gamma^k} \sum_{t=k}^{\infty} \gamma^t r_\theta(s_t, a_\phi^H, a_t^R, s_{t+1}) \right] \right] \right]$$

However, since all the actions in the first phase are communicative and thus don't impact state or reward, the first $k$ timesteps in the two phase assistance game have constant reward in expectation. Let $C = \mathbb{E}_{s_{0:k}} \left[ \sum_{t=0}^{k-1} \gamma^t r_\theta(s_t, a_\phi^H, a_{noop}^R, s_{t+1}) \right]$. This gives us:

$$\mathop{\mathbb{E}}_{\theta \sim P_\Theta, q_{0:k-1} \sim \pi_Q^{R'}, c_{0:k-1} \sim \pi^H} [ER(f'(q_{0:k-1}, c_{0:k-1}))]$$

$$= \mathop{\mathbb{E}}_{\theta \sim P_\Theta} \left[ \mathop{\mathbb{E}}_{s_0 \sim P_S, \theta \sim P_\Theta, \tau \sim \langle s_0, \theta, \pi^H, \pi^R \rangle} \left[ \frac{1}{\gamma^k} \sum_{t=0}^{\infty} \gamma^t r_\theta(s_t, a_t^H, a_t^R, s_{t+1}) \right] \right] - \frac{1}{\gamma^k} C$$

$$= \frac{1}{\gamma^k} \left( ER(\pi^R) - C \right).$$

Thus, if $\langle \pi_Q^{R'}, f' \rangle$ is a solution to the active reward learning problem, then $\pi^R$ is a solution of the two-phase communicative assistance problem. $\qquad\square$

**Corollary 5.** *If a two-phase communicative assistance problem $\langle \mathcal{M}, \pi^H, a_{noop}^R \rangle$ has exactly one communicative robot action, it can be reduced to an equivalent non-active reward learning problem.*

*Proof.* Apply Proposition 4 followed by Proposition 7. (Note that the construction from Proposition 4 does lead to an active reward learning problem with a single question, meeting the precondition for Proposition 7.) $\qquad\square$

**Proposition 2.** *Every active reward learning problem $\mathcal{P} = \langle \mathcal{M}, Q, C, \langle \Theta, r_\theta, P_\Theta \rangle, \pi^H, k \rangle$ can be reduced to an equivalent two phase communicative assistance problem $\mathcal{P}' = \langle \mathcal{M}', \pi^{H'} \rangle$.*

*Proof.* Let $\mathcal{M} = \langle S, A, \Omega, O, T, P_0, \gamma \rangle$. Let $q_0 \in Q$ be some question and $c_0 \in C$ be some (unrelated) choice. Let $N$ be a set of fresh states $\{n_0, \ldots n_{k-1}\}$: we will use these to count the number of questions asked so far. Then, we construct the new two phase communicative assistance problem $\mathcal{P}' = \langle \mathcal{M}', \pi^{H'}, a_{noop}^{R'} \rangle$ as follows:

$$\mathcal{M}' \triangleq \langle S', \{C, A^{R'}\}, \{\Omega^{H'}, \Omega^{R'}\}, \{O^{H'}, O^{R'}\}, T', P_S', \gamma, \langle \Theta, r_\theta', P_\Theta \rangle \rangle \qquad \text{Assistance game}$$

$$S' \triangleq S \cup N \qquad \text{State space}$$

$$P_S'(\hat{s}) \triangleq \mathbb{1}[\hat{s} = n_0] \qquad \text{Initial state distribution}$$

$$A^{R'} \triangleq A \cup Q \qquad \text{Robot actions}$$

$$\Omega^{H'} \triangleq S \qquad \text{H's observation space}$$

$$\Omega^{R'} \triangleq \Omega \cup N \qquad \text{R's observation space}$$

$$O^{H'}(o^{H'} \mid \hat{s}) \triangleq \mathbb{1}[o^{H'} = \hat{s}] \qquad \text{H's observation function}$$

$$O^{R'}(o^{R'} \mid \hat{s}) \triangleq \begin{cases} \mathbb{1}[o^{R'} = \hat{s}], & \hat{s} \in N \\ O(o^{R'} \mid \hat{s}, & \text{else} \end{cases} \qquad \text{R's observation function}$$

$$T'(\hat{s}' \mid \hat{s}, a^H, a^R) \triangleq \begin{cases} P_0(\hat{s}'), & \hat{s} = n_{k-1}, \\ \mathbb{1}[\hat{s}' = n_{i+1}], & \hat{s} = n_i \text{ with } i < k-1 \\ T(\hat{s}' \mid \hat{s}, a^R), & \hat{s} \in S \text{ and } a^R \in A, \\ \mathbb{1}[s' = s], & \text{else} \end{cases} \qquad \text{Transition function}$$

$$r_\theta'(\hat{s}, a^H, a^R, \hat{s}') \triangleq \begin{cases} -\infty, & \hat{s} \in N \text{ and } a^R \notin Q, \\ -\infty, & \hat{s} \in S \text{ and } a^R \in Q, \\ 0, & \hat{s} \in N \text{ and } a^R \in Q, \\ r_\theta(s, a^R, s'), & \text{else} \end{cases} \qquad \text{Reward function}$$

$$\pi^{H'}(a^H \mid o^H, a^R, \theta) \triangleq \begin{cases} \pi^H(a^H \mid a^R, \theta), & a^R \in Q \\ c_0, & \text{else} \end{cases} \qquad \text{Human policy}$$

$$a_{noop}^{R'} \triangleq q_0 \qquad \text{Distinguished noop action}$$

Technically $r_\theta'$ should not be allowed to return $-\infty$. However, since $S$ and $A$ are finite, $r_\theta$ is bounded, and so there exists some large finite negative number that is functionally equivalent to $-\infty$ that we could use instead.

Looking at the definitions, we can see $T'$ and $r'$ are independent of $a^H$, and $\pi^{H'}$ is independent of $o^H$, making this a communicative assistance problem. By inspection, we can see that every $q \in Q$ is a communicative robot action. Any $a^R \notin Q$ must not be a communicative action, because the reward $r_\theta'$ differs between $a^R$ and $q_0$. Thus, the communicative robot actions are $Q$ and the physical robot actions are $A$.

Note that by construction of $P_S'$ and $T$, we must have $s_i = n_i$ for $i \in \{0, 1, \ldots k-1\}$, after which $s_k$ is sampled from $P_0$ and all $s_t \in S$ for $t \geq k$. Given this, by inspecting $r_\theta'$, we can see that an

optimal policy must have $a_{0:k-1}^R \in Q$ and $a_{k:}^R \notin Q$ to avoid the $-\infty$ rewards. Since $a_{k:}^R \notin Q$, we have $a_{k:}^H = c_0$. Thus, setting $a_{noop}^H = c_0$, we have that the assistance problem is two phase with actions at $t_{act} = k$, as required.

Let a policy $\pi^{R'}$ for the assistance problem be **reasonable** if it never assigns positive probability to $a^R \in A$ when $t < k$ or to $a^R \in Q$ when $t \geq k$. Then, for any reasonable policy $\pi^{R'}$ we can construct an analogous pair $\langle \pi_Q^R, f \rangle$ to the original problem $\mathcal{P}$ as follows:

$$\pi_Q^R(q_i \mid q_{0:i-1}, c_{0:i-1}) \triangleq \pi^{R'}(q_i \mid o_{0:i-1}^R = n_{0:i-1}, a_{0:i-1}^R = q_{0:i-1}, a_{0:i-1}^H = c_{0:i-1}),$$

$$f(q_{0:k-1}, c_{0:k-1})(a_t \mid o_{0:t}, a_{0:t-1}) \triangleq \pi^{R'}(a_t \mid o_{0:t+k}^R, a_{0:t+k-1}^R, a_{0:t+k-1}^H),$$

where for the second equation we have

$$
\begin{aligned}
o_{0:k-1}^R &= n_{0:k-1} & a_{0:k-1}^R &= q_{0:k-1} & a_{0:k-1}^H &= c_{0:k-1} \\
o_{k:t+k}^R &= o_{0:t} & a_{k:t+k-1}^R &= a_{0:t-1} & a_{k:t+k-1}^H &= a_{noop}^H
\end{aligned}
$$

Note that this is a bijective mapping.

Consider some such policy $\pi^{R'}$ and its analogous pair $\langle \pi_Q^R, f \rangle$. By construction of $T$, we have that the first $k$ states in any trajectory are $n_{0:k-1}$ and the next state is distributed as $P_0(\cdot)$. By our assumption on $\pi^{R'}$ we know that the first $k$ robot actions must be selected from $Q$ and the remaining robot actions must be selected from $A$, which also implies (based on $\pi^H$) that after the the remaining human actions must be $c_0$. Finally, looking at $r_\theta$ we can see that the first $k$ timesteps get 0 reward. Thus:

$$
\begin{aligned}
ER_{\mathcal{P}'}(\pi^{R'}) &= \underset{s_0' \sim P_S', \theta \sim P_\theta, \tau \sim \langle s_0', \theta, \pi^{H'}, \tilde{\pi}^R \rangle}{\mathbb{E}} \left[ \sum_{t=0}^\infty \gamma^t r_\theta(s_t', a_t^{H'}, a_t^{R'}, s_{t+1}') \right] \\
&= \underset{\theta \sim P_\theta, a_{0:k-1}^{R'} \sim \pi^R, a_{0:k-1}^{H'} \sim \pi^H, s_k' \sim P_0, \tau_{k:}' \sim \langle s_k, \theta, \pi^{H'}, \tilde{\pi}^R \rangle}{\mathbb{E}} \left[ \sum_{t=k}^\infty \gamma^t r_\theta(s_t', a_t^{H'}, a_t^{R'}, s_{t+1}') \right] \\
&= \underset{\theta \sim P_\theta, q_{0:k-1} \sim \pi_Q^R, c_{0:k-1} \sim \pi^H, s_0 \sim P_0, \tau \sim \langle s_0, \theta, f(q_{0:k-1}, c_{0:k-1}) \rangle}{\mathbb{E}} \left[ \gamma^k \sum_{t=0}^\infty \gamma^t r_\theta(s_t, a_t, s_{t+1}) \right] \\
&= \gamma^k \underset{\theta \sim P_\Theta, q_{0:k-1} \sim \pi_Q^R, c_{0:k-1} \sim \pi^H}{\mathbb{E}} \left[ ER(f(q_{0:k-1}, c_{0:k-1})) \right],
\end{aligned}
$$

which is the objective of the reward learning problem scaled by $\gamma^k$.

Since we have a bijection between reasonable policies in $\mathcal{P}'$ and tuples in $\mathcal{P}$ that preserves the objectives (up to a constant), given a solution $\pi^{R*}$ to $\mathcal{P}'$ (which must be reasonable), its analogous pair $\langle \pi_Q^R, f \rangle$ must be a solution to $\mathcal{P}$. $\qquad\square$

**Corollary 3.** *Every non-active reward learning problem $\langle \mathcal{M}, C, \langle \Theta, r_\theta, P_\Theta \rangle, \pi^H, k \rangle$ can be reduced to an equivalent two phase communicative assistance problem $\langle \mathcal{M}', \pi^{H'} \rangle$.*

*Proof.* Apply Proposition 6 followed by Proposition 2. $\qquad\square$

### B.2 ASSISTANCE WITH NO REWARD INFORMATION

In a communicative assistance problem, once there is no information to be gained about $\theta$, the best thing for $R$ to do is to simply maximize expected reward according to its prior. We show this in the particular case where $\pi^H$ is independent of $\theta$ and thus cannot communicate any information about $\theta$:

**Proposition 10.** *A communicative assistance problem $\langle \mathcal{M}, \pi^H \rangle$ where $\pi^H$ is independent of $\theta$ can be reduced to a POMDP $\mathcal{M}'$ with the same state space.*

*Proof.* Given $\mathcal{M} = \langle S, \{A^H, A^R\}, \{\Omega^H, \Omega^R\}, \{O^H, O^R\}, T, P_S, \gamma, \langle \Theta, r_\theta, P_\Theta \rangle \rangle$, we define a new POMDP as $\mathcal{M}' = \langle S, A^R, \Omega^R, O^R, T', r', P_S, \gamma \rangle$, with $T'(s' \mid s, a^R) = T(s' \mid s, a_\phi^H, a^R)$ and $r'(s, a^R, s') = \mathbb{E}_{\theta \sim P_\theta} \left[ r_\theta(s, a_\phi^H, a^R, s') \right]$. Here, $a_\phi^H$ is some action in $A^H$; note that it does not

matter which action is chosen since in a communicative assistance problem human actions have no impact on $T$ and $r$.

Expanding the definition of expected reward for the assistance problem, we get:

$$\text{ER}(\pi^R) = \mathop{\mathbb{E}}_{s_0 \sim P_S, \theta \sim P_\Theta, \tau \sim \langle s_0, \theta, \pi^R \rangle} \left[ \sum_{t=0}^{\infty} \gamma^t r_\theta(s_t, a_t^H, a_t^R, s_{t+1}) \right]$$

$$= \mathop{\mathbb{E}}_{s_0 \sim P_S} \left[ \mathop{\mathbb{E}}_{\theta \sim P_\Theta} \left[ \mathop{\mathbb{E}}_{\tau \sim \langle s_0, \theta, \pi^R \rangle} \left[ \sum_{t=0}^{\infty} \gamma^t r_\theta(s_t, a_t^H, a_t^R, s_{t+1}) \right] \right] \right]$$

Note that because $\pi^H(a^H \mid o^H, a^R, \theta)$ is independent of $\theta$, the robot gains no information about $\theta$ and thus $\pi^R$ is also independent of $\theta$. This means that we have:

$$\text{ER}(\pi^R) = \mathop{\mathbb{E}}_{s_0 \sim P_S} \left[ \mathop{\mathbb{E}}_{\theta \sim P_\Theta} \left[ \mathop{\mathbb{E}}_{\tau \sim \langle s_0, \pi^R \rangle} \left[ \sum_{t=0}^{\infty} \gamma^t r_\theta(s_t, a_t^H, a_t^R, s_{t+1}) \right] \right] \right]$$

Let $r_{max} = \max_{s, a^H, a^R, s'} |r_\theta(s, a^H, a^R, s')|$ (which exists since $S$, $A^H$, and $A^R$ are finite). Then:

$$\sum_{t=0}^{\infty} \gamma^t |r_\theta(s_t, a_t^H, a_t^R, s')| \le \sum_{t=0}^{\infty} \gamma^t r_{max} = \frac{r_{max}}{1 - \gamma} < \infty.$$

So we can apply Fubini's theorem to swap the expectations and sums. Applying Fubini's theorem twice gives us:

$$\text{ER}(\pi^R) = \mathop{\mathbb{E}}_{s_0 \sim P_S} \left[ \mathop{\mathbb{E}}_{\tau \sim \langle s_0, \pi^R \rangle} \left[ \mathop{\mathbb{E}}_{\theta \sim P_\Theta} \left[ \sum_{t=0}^{\infty} \gamma^t r_\theta(s_t, a_t^H, a_t^R, s_{t+1}) \right] \right] \right]$$

$$= \mathop{\mathbb{E}}_{s_0 \sim P_S} \left[ \mathop{\mathbb{E}}_{\tau \sim \langle s_0, \pi^R \rangle} \left[ \sum_{t=0}^{\infty} \gamma^t \mathop{\mathbb{E}}_{\theta \sim P_\Theta} \left[ r_\theta(s_t, a_t^H, a_t^R, s_{t+1}) \right] \right] \right]$$

$$= \mathop{\mathbb{E}}_{s_0 \sim P_S} \left[ \mathop{\mathbb{E}}_{\tau \sim \langle s_0, \pi^R \rangle} \left[ \sum_{t=0}^{\infty} \gamma^t r'(s_t, a_t^R, s_{t+1}) \right] \right].$$

In addition, the trajectories are independent of $\pi^H$, since the assistance problem is communicative, and so for a given policy $\pi^R$, the trajectory distributions for $\mathcal{M}$ and $\mathcal{M}'$ coincide, and thus the expected rewards for $\pi^R$ also coincide. Thus, the optimal policies must coincide. $\square$

## C   EXPERIMENTAL DETAILS

### C.1   PLANS CONDITIONAL ON FUTURE FEEDBACK

In the environment described in Section 4.1, $R$ needs to bake either apple or blueberry pie (cherry is never preferred over apple) within 6 timesteps, and may query $H$ about her preferences about the pie. Making the pie takes 3 timesteps: first $R$ must make flour into dough, then it must add one of the fillings, and finally it must bake the pie. Baking the correct pie results in +2 reward, while baking the wrong one results in a penalty of -1. In addition, $H$ might be away for several timesteps at the start of the episode. Querying $H$ costs 0.1 when she is present and 3 when she is away.

The optimal policy for this environment depends on whether $H$ would be home early enough for $R$ to query her and bake the desired the pie by the end of the episode. $R$ should always quickly make dough, as that is always required. If $H$ returns home on timestep 4 or earlier, $R$ should wait for her to get home, ask her about her preferences and then finish the desired pie. If $H$ returns home later, $R$ should make its best guess about what she wants, and ensure that there is a pie ready for her to eat: querying $H$ when she is away is too costly, and there is not enough time to wait for $H$, query her, put in the right filling, and bake the pie.

We use PBVI to train an agent for this assistance problem with different settings for how long $H$ is initially away. As expected, this results in a policy that makes dough, queries $H$ and bakes the correct pie if $H$ is back on timestep 4 or earlier; if $H$ is back on timestep 5 or 6, $R$ simply makes apple pie as that is most likely to be what $H$ wants.

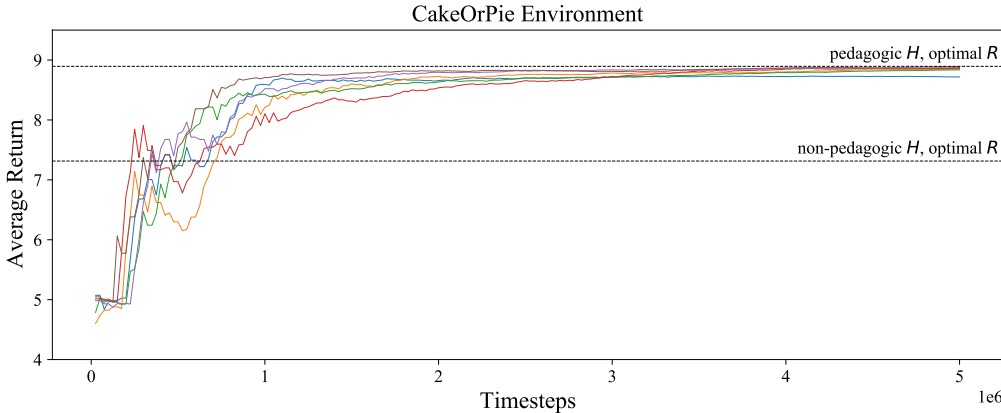

Figure 4: DQN smoothed learning curves on the CakeOrPie environment, with 6 seeds over $5M$ timesteps and learning rate of $10^{-4}$.

## C.2 RELEVANCE-AWARE ACTIVE LEARNING: OPTIMAL GRIDWORLDS

In the wormy-apple environment described in Section 4.2, the robot had to bring the human some apples in order to make a pie, but there's a $20\%$ chance that the apples have worms in them, and the robot does not yet know how to dispose of soiled apples. The robot gets 2 reward for making an apple pie (regardless of how it disposed of any wormy apples), and gets $-2$ reward if it disposes of the apples in the wrong container. Additionally, asking a question incurs a cost of $0.1$. We solve this environment with exact value iteration.

If the environment is two-phase, with a lower discount rate ($\lambda = 0.9$), $R$'s policy never asks questions and instead simply tries to make the apple pie, guessing which bin to dispose of wormy apples in if it encounters any. Intuitively, since it would have to always ask the question at the beginning, it would always incur a cost of $0.1$ as well as delay the pie by a timestep resulting in 10% less value, and this is only valuable when there turn out to be worms *and* its guess about which bin to dispose of them in is incorrect, which only happens 10% of the time. This ultimately isn't worthwhile. This achieves an expected undiscounted reward of $1.8$. Removing the two-phase restriction causes $R$ to ask questions mid-trajectory, even with this low discount. With this result achieves the maximal expected undiscounted reward of $1.98$.

With a higher discount rate of $\lambda = 0.99$, the two-phase policy will always ask about which bin to dispose of wormy apples in, achieving $1.9$ expected undiscounted reward. This is still less than the policy without the two-phase restriction, which continues to get undiscounted reward $1.98$ because it avoids asking a question 80% of the time, and so incurs the cost of asking a question less often.

## C.3 LEARNING FROM PHYSICAL ACTIONS: CAKE-OR-PIE EXPERIMENT

In the environment described in Section 4.3, $H$ wants a dessert, but $R$ is unsure whether $H$ prefers cake or pie. Preparing the more desired recipe provides a base value of $V = 10$, and the less desired recipe provides a base value of $V = 1$. Since $H$ doesn't want the preparation to take too long, the actual reward when a dessert is made is given by $r_t = V \cdot f(t)$, with $f(t) = 1 - (t/N)^4$, and $N = 20$ as the episode horizon.

The experiments use the pedagogic $H$, that picks the chocolate first if they want cake, which allows $R$ to distinguish the desired recipe early on - this is in contrast with the non-pedagogic $H$, which does not account for $R$ beliefs and always goes for the dough first.

With the pedagogic $H$, the optimal $R$ does not move until $H$ picks or skips the dough; if $H$ skips the dough, this implies the recipe is cake and $R$ takes the sugar, and then the cherries - otherwise it goes directly for the cherries. With the non-pedagogic $H$, the optimal $R$ goes for the cherries first (since it is a common ingredient), and only then it checks whether $H$ went for the chocolate or not, and has to go all the way back to grab the sugar if $H$ got the chocolate.

We train $R$ with Deep Q-Networks (DQN; (Mnih et al., 2013)); we ran 6 seeds for $5M$ timesteps and a learning rate of $10^{-4}$; results are shown in Figure 4.

## D  OPTION VALUE PRESERVATION

In Section 4.1, we showed that $R$ takes actions that are robustly good given its uncertainty over $\theta$, but waits on actions whose reward will be clarified by future information about $\theta$. Effectively, $R$ is preserving its *option value*: it ensures that it remains capable of achieving any of the plausible reward functions it is uncertain over.

A related notion is that of *conservative agency* (Turner et al., 2020), which itself aims to preserve an agent's ability to optimize a wide variety of reward functions. This is achieved via *attainable utility preservation* (AUP). Given an agent optimizing a reward $r_{\text{spec}}$ and a distribution over auxiliary reward functions $r_{\text{aux}}$, the AUP agent instead optimizes the reward

$$r_{AUP}(s,a) = r_{\text{spec}}(s,a) - \lambda \mathop{\mathbb{E}}_{r_{\text{aux}}} \left[\max(Q_{r_{\text{aux}}}(s, a_\phi) - Q_{r_{\text{aux}}}(s,a), 0)\right]$$

where the hyperparameter $\lambda$ determines how much to penalize an action for destroying option value, and $a_\phi$ is an action that corresponds to $R$ "doing nothing".

However, the existing AUP penalty is applied to the *reward*, which means it penalizes any action that is part of a long-term plan that destroys option value, even if the action itself does not destroy option value. For example, in the original Kitchen environment of Figure 1 with a sufficiently high $\lambda$, any trajectory that ends with baking a pie destroys option value and so would have negative reward. As a result, there is no incentive to make dough: the only reason to make dough is to eventually make a pie, but we have established that the value of making a pie is negative.

What we need is to only penalize an action when it is going to *immediately* destroy option value. This can be done by applying the penalty during *action selection*, rather than directly to the reward:

$$\pi_{AUP}(s) = \operatorname*{argmax}_a Q_{r_{\text{spec}}}(s,a) - \lambda \mathop{\mathbb{E}}_{r_{\text{aux}}} \left[\max(Q_{r_{\text{aux}}}(s, a_\phi) - Q_{r_{\text{aux}}}(s,a), 0)\right]$$

After this modification, the agent will correctly make dough, and stop since it does not know what filling to use.

In an assistance problem, $R$ will only preserve option value if it expects to get information that will resolve its uncertainty later: otherwise, it might as well get what reward it can given its uncertainty. Thus, we might expect to recover existing notions of option value preservation in the case where the agent is initially uncertain over $\theta$, but will soon learn the true $\theta$. Concretely, let us consider a fully observable communicative Assistance POMDP where the human will reveal $\theta$ on their next action. In that case, $R$'s chosen action $a$ gets immediate reward $\hat{r}(s,a) = \mathbb{E}_\theta\left[r_\theta(s,a)\right]$, and future reward $\mathbb{E}_{\theta \sim P_\Theta, s' \sim T(\cdot|s,a)}\left[V_\theta(s')\right]$, where $V_\theta(s)$ refers to the value of the optimal policy when the reward is known to be $r_\theta$ and the initial state is $s$. Thus, the agent should choose actions according to:

$$\operatorname*{argmax}_a \mathop{\mathbb{E}}_{s' \sim T(\cdot|s,a)} \left[\mathop{\mathbb{E}}_\theta\left[r_\theta(s,a) + \gamma V_\theta(s')\right]\right]$$

$$= \operatorname*{argmax}_a \mathop{\mathbb{E}}_{s' \sim T(\cdot|s,a)} \left[\hat{r}(s,a) + \gamma V_{\hat{r}}(s') - \gamma V_{\hat{r}}(s') + \gamma \mathop{\mathbb{E}}_\theta\left[V_\theta(s')\right]\right]$$

$$= \operatorname*{argmax}_a Q_{\hat{r}}(s,a) - \gamma \mathop{\mathbb{E}}_\theta \left[\mathop{\mathbb{E}}_{s' \sim T(\cdot|s,a)}\left[V_{\hat{r}}(s')\right] - \mathop{\mathbb{E}}_{s' \sim T(\cdot|s,a)}\left[V_\theta(s')\right]\right]$$

This bears many resemblances to the AUP policy, once we set the distribution over auxiliary rewards to be the distribution over $r_\theta$, along with $r_{\text{spec}} = \hat{r}$ and $\lambda = \gamma$. Nonetheless, there are significant differences, primarily because AUP was designed for the case where $r_{\text{spec}}$ and $r_{\text{aux}}$ could be arbitrarily different, which is not the case for us. In particular, with AUP the agent is penalized for any loss in $r_{\text{aux}}$ by taking the chosen action $a$ *relative to doing nothing*, while in the assistance problem, the agent is penalized for any loss in $r_\theta$ by acting according to $\hat{r}$ *relative to what could be achieved if $R$ knew the true reward*. It is intriguing that both these methods lead to behavior that we would characterize as "preserving option value".

