# OpenReview forum: "Benefits of Assistance over Reward Learning"
_ICLR.cc/2021/Conference — Reject_

### Official Review · AnonReviewer3 · 2020-10-15
**It seems to me that the authors disregard a rich related literature**

**Rating:** 5
**Confidence:** 2

**Review:**

------------------------------------------------------------
Post-rebuttal
-----------------------------------------------------------
Given the effort of the authors of improving their manuscript, I am improving my original score. However, my evaluation is still "weak reject" for the reasons below:

(1) I still fail to see clear differences between "assistance", as defined by the authors, and the other reinforcement learning-like approaches that assume that the reward function is unknown. I can see that they perhaps provide a more organized and methodological description of how that "assistance" can happen when compared to the previous works. However, the paper lacks practical advice, exactly how should I build an agent to leverage such "assistance"? I don't think their ideas are so novel that other methods couldn't be at least adapted to work in their scenario (to include some empirical evaluation in the manuscript).

(2) The paper seems a little displaced to me in this conference. The paper neither provides practical and direct guidance on how to build algorithms to leverage "assistance", nor is a survey that focuses on organizing the area and discussing differences between works. Perhaps the paper would be better placed in a "Blue Sky" track.

-----------------------------------------------------------

The authors propose two learning paradigms where the learning agent doesn't have access to a reward function but instead has to learn directly from the "assistance" from a trainer agent.

While looking for ways to facilitate task specification and human integration in the learning process is a relevant and promising research goal, the authors don't explain the difference between their newly-proposed paradigms and the very rich literature on Inverse Reinforcement Learning and Learning from Demonstrations.

Learning from human "assistance" instead of a reward function is not a new thing, and the surveys below (not cited by the authors) summarize a rich literature that does precisely that:

Silva, Felipe Leno, and Anna Helena Reali Costa. "A survey on transfer learning for multiagent reinforcement learning systems." Journal of Artificial Intelligence Research 64 (2019): 645-703. -> Surveys many categories of works where one agent provides guidance to others, including humans providing "assistance" to learning agents.

Argall, Brenna D., et al. "A survey of robot learning from demonstration." Robotics and autonomous systems 57.5 (2009): 469-483. -> Surveys learning from demonstration, where a human provides policy demonstrations to a learning agent, which usually doesn't have access to a reward function

Gao, Yang, et al. "A survey of inverse reinforcement learning techniques." International Journal of Intelligent Computing and Cybernetics (2012). -> Inverse reinforcement learning, where the learning agent doesn't have access to a reward function and has to infer a policy from human assistance

Without a comprehensive discussion about the differences between the authors' proposal and those paradigms, I can't judge the paper contribution. To my eyes, all the descriptions gave in the paper look just like the same as one of the problems surveyed in the papers above.

For example, "Non-active reward learning" seems to me equivalent to inverse reinforcement learning.

Also, "active reward learning" seems to me a special case of the action advising problem surveyed in the first paper above.

In the case the authors deal with a different problem in this paper, I suggest that the manuscript is rewritten to thoughtfully explain the difference between all those scenarios. In case they indeed are correspondent scenarios, I suggest that the authors use the same notation as the previous works, and include comparisons with the state of the art methods in the experimental evaluation.

-----------------------------------------------------
Other suggestions

- I don't get how the policy decision function will compute the expected reward if the reward function is unknown.

- You assume access to a "human decision function", what exactly does that mean? do you need the probabilities for taking each action? If that's the case it is very unrealistic to expect that a human is able to provide probabilities for each action. Asking s/he to simply pick one action when requested is more realistic.

---

> ### Author Response · Authors · 2020-11-24
> **Yes, reward learning includes the papers you cite**
>
> Thanks for the review. It sounds like your primary concern is that there is existing literature on inverse reinforcement learning and learning from demonstrations, which is very similar to our reward learning paradigm (whether active or non-active).
>
> You are correct that the reward learning paradigm is very similar to these existing areas. We explain this in Section 2.2, where we cited Ng et al 2000, Ziebart et al 2010, and Fu et al 2017, which are all canonical inverse reinforcement learning papers. In fact, it also includes a large variety of other work, such as learning from preferences, for which we cited Zhang et al 2017 Wirth et al 2017, Christiano et al 2017, and Sadigh et al 2017. There are of course many other papers on all of these topics, and we could not cite them all; this is not intended to be a survey paper. We have added citations to the surveys you mention, as we expect those will be useful to readers.
>
> This existing literature is exactly why we wrote this paper: almost all of the existing literature on learning without reward functions can be captured by the reward learning paradigm as we formalize it. But (as we show) the assistance paradigm can enable significantly better behavior from the agents we train! We are hoping to influence researchers to put more effort into algorithms for the assistance domain, in order to realize these qualitative benefits, instead of continuing to work in the reward learning paradigm as they have done so far.
>
> We would appreciate suggestions on how we can make this clearer in the paper, as it is crucial to our main point.
>
> ----
>
> Responses to other points:
>
> > I don't get how the policy decision function will compute the expected reward if the reward function is unknown.
>
> We’re not totally sure what you’re asking, so we’ll try two different explanations:
>
> Explanation 1: The policy decision function can take an expectation over the form of the reward, and then take an expectation over the trajectory that the policy will take, in order to get the expected reward of a given policy.
>
> Explanation 2: Consider Atari games. From the perspective of an AI agent, there is only a set of pixels; there’s no obvious or trivial way of computing the reward from these pixels. However, the _environment_ does not just consist of pixels -- it also includes the memory in the Atari emulator. Given this, the reward is easy to calculate -- we just extract the number in the “score” variable in the memory. Similarly, our _environments_ know the reward function, but the _agent_ does not.
>
> > You assume access to a "human decision function", what exactly does that mean? do you need the probabilities for taking each action? If that's the case it is very unrealistic to expect that a human is able to provide probabilities for each action. Asking s/he to simply pick one action when requested is more realistic.
>
> If we are training with a human-in-the-loop, the human is only queried when stepping the environment forward. As a result, in this setting we only need the human to choose a particular action, rather than give probabilities over each action.
>
> However, we expect it will be more typical to train with a simulated human, which need not be perfectly accurate, especially if we use deep reinforcement learning. This simulated human can take the same form as the human models used in inverse reinforcement learning. For example, a common choice of human model is the Boltzmann-rational model, in which we have
>
> $$\pi^H(a \mid s, \theta) \propto \exp(Q(s, a; \theta))$$
>
> That is, the human is modeled as being more likely to choose actions that are higher-value. Notice that given a specific $\theta$, the probability of a particular action can be computed fully automatically, without asking a real human for any input.

---

> > ### Comment · AnonReviewer3 · 2020-11-24
> > **Contribution**
> >
> > It is still unclear to me the difference between the "reward learning paradigm" and the "assistance paradigm".
> >
> > As you say "almost all of the existing literature on learning without reward functions can be captured by the reward learning paradigm as we formalize it", I understand that there isn't any difference and that all the approaches in which the agent learns without explicitly knowing the reward function are contained in the most "general class" of methods named by you as "assistance paradigm".
> >
> > In this case, what is the contribution of the paper? Defining a unified framework seems to me a good contribution to a survey, which the own authors state that this is not the case for this specific paper.
> >
> > For a non-survey paper, I would expect to extract a specific empirical method from the paper (which would require comparisons against state-of-the-art similar methods), or new discoveries enabled through theoretical analysis. I can't clearly see any of those.

---

> > > ### Author Response · Authors · 2020-11-24
> > > **Contribution is the identification of qualitative behaviors and the restrictions under which assistance is equivalent to reward learning**
> > >
> > > The contributions are:
> > >
> > > 1. Precisely identifying the differences between assistance and reward learning, by showing that the communicative and two phase restrictions together make assistance equivalent to reward learning.
> > > 2. Showing that these differences matter: by removing the restrictions, we get several new qualitative behaviors, described in Section 4.
> > >
> > > > For a non-survey paper, I would expect to extract a specific empirical method from the paper (which would require comparisons against state-of-the-art similar methods), or new discoveries enabled through theoretical analysis. I can't clearly see any of those.
> > >
> > > We argue that the qualitative behaviors in assistance (Section 4) are a "new discovery" and thus are an important contribution even though they do not take the form of theorems. (Arguably, they were enabled by theoretical analysis; we generated them by asking ourselves what useful behaviors could not be expressed by a two phase algorithm; the concept of "two phase" came about because of theoretical analysis.)

---

### Official Review · AnonReviewer1 · 2020-10-28

**Rating:** 4
**Confidence:** 3

**Review:**

Summary:
This work proposes learning a single control policy for human-in-the-loop learning rather than having a reward learning component and a control component. The key difference is that the action selection can use information from the reward learning module. The authors formulate an assistance game in this setting and show that it can reduce to an equivalent POMDP. The work then describes a communicative assistance problem and shows the equivalence of reward learning to assistance and visa versa. Results show qualitative improvements on variants of the kitchen domain.

Pros:
- The paper was overall well-written (although some key differences with prior work were unclear, as described below).
- The high-level area of human-in-the-learning is a useful and important space.

Cons:
- There are many works in human-in-the-loop learning, and it wasn’t clear whether the ideas in this paper were novel enough at a high-level. Some relevant works are included below.
- The paper was hard to follow with respect to the distinction between reward learning and assistance. Adding more examples or describing the difference from other angles could be useful.
- The experimental domains were simple and there were no computational results shown in the main paper. This is important to show in order to support the claims of the paper.

Comments:
- The qualitative results are useful for showing specific cases in which the proposed approach may be beneficial but this restricts how applicable the approach seems to be. It would be nice to see computational results over a variety of domains and comparisons with baseline approaches to see the benefit of the approach more generally.
- Section 4.2 focuses on asking the right questions at the right time. What is the main novelty here with respect to more general active learning approaches?
- A few minor notational confusions: for example, it’s confusing to have R be the robot as it’s often the reward function. And in Section 4.1, C is used for choices and for cherry.
- Other work that might be relevant to the problem proposed here:

Game-theoretic modeling of human adaptation in human-robot collaboration
S Nikolaidis, S Nath, AD Procaccia, S Srinivasa

Maximizing BCI Human Feedback using Active Learning
Z Wang, J Shi, I Akinola, P Allen

Modeling humans as observation providers using pomdps
S Rosenthal, M Veloso

Active Learning for Risk-Sensitive Inverse Reinforcement Learning
R Chen, W Wang, Z Zhao, D Zhao

Contact: Deciding to communicate during time-critical collaborative tasks in unknown, deterministic domains
VV Unhelkar, JA Shah

Efficient model learning from joint-action demonstrations for human-robot collaborative tasks
S Nikolaidis, R Ramakrishnan, K Gu, J Shah

Recommendation:
Overall, I thought the work at a high-level was limited in its novelty compared with prior work in human-in-the-loop learning. The distinction between reward learning and assistance, a key part of the paper, was hard to fully understand so clarifying this description through examples or more clear text would be valuable. The evaluation was also simple and not very convincing with respect to supporting the claims. Adding computational experiments and appropriate baselines would be important to show the general applicability of the approach.

--------------------
Response after rebuttal:
Thank you to the authors for their response. I appreciate the detailed answers to each of the prior works. I still do think the paper needs to be more clear in the problem and differentiation with prior work in the writing itself, which will require a non-trivial update to the paper.

On the computational results + baselines side, while the authors have run the experiment and have described these qualitative behaviors, this isn't a substitute for quantitative results, especially because this is important to show when comparing with baselines. The authors say "We have updated Section 4 to be clearer about what baseline approaches would do in the environments we have tested". It's important to show evidence that this is what the baselines actually did.

Based on these points, I don't think the paper is quite ready for publication yet.

---

> ### Author Response · Authors · 2020-11-24
> **The cited papers do not show the qualitative behaviors we identify**
>
> Thanks for the review, and the links to related work! We’re glad that you found the topic important and the paper well-written. We hope to convince you that the qualitative behaviors we illustrate are indeed novel.
>
> > There are many works in human-in-the-loop learning, and it wasn’t clear whether the ideas in this paper were novel enough at a high-level.
>
> All of these works either:
> 1. Have a known reward function, or
> 2. Fall within the reward learning paradigm, or
> 3. Are related to the assistance paradigm but don’t illustrate the qualitative behaviors we do.
>
> We provide more details on each of the papers below:
>
> > Game-theoretic modeling of human adaptation in human-robot collaboration
>
> This paper assumes that the robot knows the reward function and focuses on the robot communicating to the human, whereas we focus on cases where the robot does not know what must be done and the robot must learn from the human.
>
> > Maximizing BCI Human Feedback using Active Learning
>
> This paper is a central example of the active reward learning paradigm, for which we have several other citations.
>
> > Modeling humans as observation providers using pomdps
>
> This paper also assumes a known reward function. That being said, there is an equivalence between unknown rewards and unknown observations, so this paper is related -- in particular, their HOP-POMDPs can be used to encode communicative assistance problems, and so can encode the environments of Sections 4.1 and 4.2 (though not Section 4.3). However, the paper does not describe any of the qualitative behaviors of Section 4, nor does it identify the differences with reward learning, and so it is mostly orthogonal to our paper. We have mentioned this in Section 3 when introducing communicative assistance problems.
>
> > Active Learning for Risk-Sensitive Inverse Reinforcement Learning
>
> This is another central example of the active reward learning paradigm.
>
> > Contact: Deciding to communicate during time-critical collaborative tasks in unknown, deterministic domains
>
> This seems to be a paper about multiagent communication? As far as we can tell it cannot be applied to human-in-the-loop learning, because it assumes that we can control the policies of all of the agents, but we cannot control the human’s policy. (Perhaps we are misunderstanding the application of the paper to human-AI interaction though.)
>
> > Efficient model learning from joint-action demonstrations for human-robot collaborative tasks
>
> This is probably the most related paper -- it does have a robot that assists a human, and it uses POMDP-style reasoning to make inferences about what the human prefers. It is best thought of as an approach to solve an assistance problem when there are joint human-robot demonstrations on which the robot can be trained. We have added a sentence about this paper in Section 2.3.
>
> We emphasize that our contribution is *not* in the definition of the assistance formalism, which has been done in prior work: our contribution is in the comparison to reward learning and the illustration of the qualitative benefits of assistance over reward learning.
>
> > Section 4.2 focuses on asking the right questions at the right time. What is the main novelty here with respect to more general active learning approaches?
>
> Can you provide citations for an active learning approach that would show the behavior we illustrate in Section 4.2? All the approaches we know of would either ask about wormy apples at the beginning (rather than after picking up an apple with worms), or would never ask about wormy apples.
>
> > The experimental domains were simple and there were no computational results shown in the main paper. This is important to show in order to support the claims of the paper.
>
> We have implemented the environments of Section 4, and run experiments that show the qualitative behaviors that we describe. Does that address your comment? If not, what additional results would you want, and why do you think they are necessary to support our claims?
>
> > It would be nice to see computational results over a variety of domains and comparisons with baseline approaches to see the benefit of the approach more generally.
>
> We have updated Section 4 to be clearer about what baseline approaches would do in the environments we have tested, and added an interactive reward learning baseline. Unfortunately, we don’t know of environment suites that have been designed with assistance in mind, and so we cannot test the approach in multiple domains without building these environments ourselves, which would require a huge effort.
>
> > A few minor notational confusions: for example, it’s confusing to have R be the robot as it’s often the reward function. And in Section 4.1, C is used for choices and for cherry.
>
> Good point, we will think about how to change these. We did think about R and didn’t find any alternatives we liked, but we hadn’t noticed the conflict with C.

---

### Official Review · AnonReviewer2 · 2020-10-28
**[UPDATED] [Official Review]: Well written, but a bit too one-sided**

**Rating:** 7
**Confidence:** 4

**Review:**

UPDATE:
After extensive discussion with the authors, I'm raising my score to a 7.
I believe their revision will adequately address the concerns I've raised.
I think this paper clearly identifies and illustrates the qualitative advantages of assistance, and that this is a novel and significant contribution.

In particular, I do *not* believe, as other reviewers seem to, that any of the following are sufficient reasons for rejecting this work:
* The wealth of prior work on variants of reward learning and assistance
* The lack of a comprehensive survey or categorization of such work in this submission
* The lack of further results

After reading the other reviews and responses, I am more confident that this paper makes a valuable contribution, although I stand ready to be challenged by other reviewers.  This is because the authors have argued that the qualitative benefits they describe in sections 4.1/2/3 have not been available to any of the many previous works reviewers mentioned, and no reviewers disputed this.  Furthermore, I did not find the reasons provided for rejection to be very relevant to the goals of this work.  So overall, I do not believe that other reviewers have made a strong case for rejecting this work.

In my mind, the best argument would seem to be simply that the contribution is insufficient.  I think this is a common criticism of papers that do not adhere to a conventional format or "type", but in this case, it seems unfair.  I believe the intellectual contribution of this paper is rather modest, but nonetheless novel and significant.  And I found this motivation for the work quite compelling (emphasis mine):
> This existing literature is exactly why we wrote this paper: almost all of the existing literature on learning without reward functions can be captured by the reward learning paradigm as we formalize it. But (as we show) the assistance paradigm can enable significantly better behavior from the agents we train! **We are hoping to influence researchers to put more effort into algorithms for the assistance domain, in order to realize these qualitative benefits, instead of continuing to work in the reward learning paradigm as they have done so far.**

I would encourage the authors to explain this goal in their revision, and make sure their claims about the superiority of assistance are appropriately modest.  Overall, I think the qualitative benefits of assistance presented provide a compelling argument for more work in the assistance paradigm, *given the paucity of such work*.  But I think the overall message of the paper should be: "Given these advantages, and the lack of work on assistance, there should be more work on assistance, since it seems promising and neglected", and not "Assistance is better, so why would you do reward learning?"  And my first impression of the paper was closer to the latter.

END UPDATE

-------------------------
Evaluation:

This paper is well written and makes a nice point regarding qualitative advantages of assistance over reward learning.  Specifically, the authors show how assistance naturally leads an agent to "agent to (1) choose questions based on their relevance, (2) create plans whose success depends on future feedback, and (3) learn from physical human actions in addition to communicative feedback."  However, the framing is quite one-sided, and the authors make no mention of potential advantages of reward learning.  And it is possible to achieve the same qualitative advantages by slightly modifying the (rather restrictive) formulation of reward learning that the authors use.  I think the work should either include such a discussion, or offer more convincing evidence regarding when assistance is in fact preferable in practice (e.g. experiments with some features that might advantage either approach).

Some potential advantages of reward learning are:
- reduced complexity of the learning problem
- the human retains more control
- it seems to require less modeling of human psychology
- less opportunities to corrupt the reward signal

Regarding the qualitative advantages mentioned in Sections 4.1/2, we can achieve the same benefits in many instances by incorporating regular feedback sessions where H and R can communicate.  I was confused by the claimed advantage in 4.3, since IRL already can learn from physical actions.

Overall, I think this paper sets up a bit of a false dichotomy between (non-interactive) reward learning, and the full assistance game formulation.  We can instead view these methods as varying wrt 1) when and how we ask R to interpret H's actions as communicative, and 2) how interactive the learning process is.
This paper still makes important (corresponding) points: 1) there is a benefit to considering H's behavior as communicative whenever we understand its semantics, and 2) interaction is important.

These are not very surprising, although very nicely and clearly argued for and demonstrated. However, regarding point (1): if we don't understand the semantics of H's behavior, it makes sense to restrict the communication to a more well-defined channel, as in reward learning; furthermore, even if we *do* know the semantics, H may not wish all of her behavior to be viewed as communicative, and allowing her to directly control when and how her behavior will be viewed as communicative grants H more agency over the behavior of R.

Accepting points (1) and (2), it still seems like most effective methods are likely to lie somewhere between the two extremes described in this work.  For example, the authors state: "Thus, for good performance we need to model π^H. This will require insights from a broad range of fields that study human behavior."  However, accurately modelling π^H  seems significantly less urgent when communication from H is more limited and literal (i.e. when we move closer towards current reward learning practice).  As another piece of loosely supporting evidence, in "Literal or Pedagogic Human? Analyzing Human Model Misspecification in Objective Learning", Milli and Dragan show that more sophisticated assumptions about the semantics of human behaviors may also be more brittle.

Another contribution of the paper is to explicitly frame reward learning as a special case of assistance.  This seems like a straightforward, minor, contribution.

I was also disappointed that the paper didn't discuss reward corruption. The need to more fully understand H in the assistance paradigm creates more opportunities for R to misinterpret H's behavior.  In practice, the tighter feedback loops favored by the assistance paradigm might also create more opportunities for R to (e.g. irreversibly) "corrupt" H's reward function and/or policy, since an initial misunderstanding could be self-reinforcing.  For instance, H might fail to tell R that its behavior was intimidating if doing so previously had led R to become more (instead of less) intimidating.  This could lead R to become confident that this behavior was *not* intimidating to H.  Such a scenario seems less likely when H provides feedback to R in an "offline" or "purely communicative" context.

-------------------------
Detailed Suggestions:
- Figure 1 caption could be clearer (what is depicted on the right?)
- "These behaviors cannot be expressed by a reward learning agent." <-- not literally true; suggest a rephrase.
- "Since c0:k-1 only serves to provide information to R" <-- It doesn't have to... e.g. people do IRL on data that doesn't fit this description.
- Provide more of an informal introduction in section 2.3 before jumping into definitions
- "However, since we want to compare to reward learning, we follow its assumptions and so assume that the human policy πH is available." While I was eventually able to understand this reasoning, it sounds false out of context, and should be clarified.
- a^H_noop is not defined
- I consider the first paragraph of Section 4 to be the "meat" of the paper, and I think some of this content should be front-loaded more.  It's a shame to wait until page 6.
- I would start each of 4.1/2/3 with a paragraph giving a non-mathematical, qualitative description of the result.
- I like the paragraph at top of page 7, which gives an explanation of how reward learning would act/fail.  Can you include such a paragraph in 4.1/3 as well?  I think the more you mirror structure in these sections, the better.

---

> ### Author Response · Authors · 2020-11-24
> **Responses to detailed suggestions (can be ignored)**
>
> > Detailed Suggestions
>
> Thanks for these! We’ve updated the paper accordingly, and expect these updates will significantly increase the clarity of the paper. Some notes on specific suggestions below:
>
> > "These behaviors cannot be expressed by a reward learning agent." <-- not literally true; suggest a rephrase.
>
> We’ve added a parenthetical “(as formalized in this paper)”, but we believe it is literally true as written. The behaviors in Sections 4.1 and 4.2 require an agent that can act, then learn, and then act again; whereas reward learning (as we formalize it) requires only two phases. The behavior in Section 4.3 requires the agent to learn from physical actions and then act _within a single episode_; this cannot even be expressed in the reward learning paradigm (as we formalize it).
>
> > "Since c0:k-1 only serves to provide information to R" <-- It doesn't have to... e.g. people do IRL on data that doesn't fit this description.
>
> The intended meaning was that _within the reward learning problem_ (what R has to do) the choices can only provide information to R, they don’t somehow help R to gain expected reward (because they don’t affect the environment that R acts in). We did not mean to imply that $c_{0:k-1}$ weren’t used for some other purpose previously; just that they don’t affect what R has to do now. We have rephrased to “Since $H$'s choices $c_{0:k-1}$ do not affect the state of the environment that $R$ is acting in”.
>
> > a^H_noop is not defined
>
> The first mention is in the equation:
>
> $\exists a^H_{noop} \in A^H, \forall \tau \in \text{Traj}(\pi^{R*}), \; \left[ \forall t < t_{act} : a^R_t\text{ is communicative }\wedge \forall t \geq t_{act} : \; a^H_t = a^H_{noop} \right]$
>
> $a^H_{noop}$ is simply whichever human action makes this equation hold (it is bound by the initial $\exists$ quantifier.) Perhaps the name is too evocative though -- do you have suggestions for alternative notation here?
>
> > I consider the first paragraph of Section 4 to be the "meat" of the paper, and I think some of this content should be front-loaded more.  It's a shame to wait until page 6.
>
> Indeed, we would also have liked to do this. The main issue is that the qualitative examples in Section 4 do depend on having adequate formalizations (Section 2) as well as the notions of “communicative” and “two phase” (Section 3), and so it seemed quite challenging to bring Section 4 any earlier in the paper. If there was no space limitation, we could consider having an extra section between Sections 1 and 2 that provide the qualitative examples, say what behavior we want from them, and say why reward learning wouldn’t show those behaviors, but then we would still need to have the current Section 4 later to explain how exactly we do get those behaviors from the assistance paradigm, which means this would lengthen the paper too much.
>
> Figure 1 is our attempt to front-load some of the content of Section 4; we’d be excited to hear any other suggestions you have for front-loading more of the content.
>
> > I would start each of 4.1/2/3 with a paragraph giving a non-mathematical, qualitative description of the result.
>
> We thought we already had some of this (though not in 4.1), but we’ve added some more details. We’re curious if this addresses what you were thinking about, and if not what you would suggest instead.
>
> > I like the paragraph at top of page 7, which gives an explanation of how reward learning would act/fail.  Can you include such a paragraph in 4.1/3 as well?
>
> Yeah, in hindsight this is a clear improvement. We’ve done this for both reward learning and an interactive variant of reward learning. Thanks!

---

> > ### Comment · AnonReviewer2 · 2020-11-24
> > **response**
> >
> > > "These behaviors cannot be expressed by a reward learning agent."
> >
> > I maintain that this claim is literally false, and should be removed or qualified.
> > See "Scalable agent alignment via reward modeling: a research direction" Leike et al. (2018), bottom of page 6 for an (informal) proof that *any* behavior can be expressed via a reward function.  Manually specifying the correct reward function is a degenerate form of reward modelling (with a prior that's a delta function on the specified reward function), but the general point is that a (e.g. learned) reward function can, in principle, be appropriately responsive to arbitrary kinds of feedback/interaction with a human.
> > By the way, you should probably cite this work; the only reason not to in my mind is that it's only a tech report.
> >
> > > Figure 1 is our attempt to front-load some of the content of Section 4; we’d be excited to hear any other suggestions you have for front-loading more of the content.
> >
> > I didn't have specific things in mind.  But I think you could expand and improve Figure 1.  For instance, the first sentence of the caption could say: "An illustration of the qualitative behaviors that assistance, but not reward learning, encourages" (or something like that).  And the figure should also have a title above the 4 robots, I think, (e.g. "benefits of assistance").  I would recommend spending some significant effort on improving this figure, and try to make it easily stand on its own.  Show it to people who haven't read the paper, etc.
> >
> > The paragraph that begins "Consider for example the kitchen environment illustrated in Figure 1" could also begin instead by saying something like: "In our paper, we show that assistance leads to qualitatively different (advantageous) behaviors, as illustrated in Figure 1.  And then you could also make this paragraph a list.  Overall, it wasn't blindingly obvious from this paragraph that these were the meat of the paper, so I'd work on that.

---

> > > ### Author Response · Authors · 2020-11-25
> > > **Thanks**
> > >
> > > > I maintain that this claim is literally false, and should be removed or qualified. See "Scalable agent alignment via reward modeling: a research direction" Leike et al. (2018), bottom of page 6 for an (informal) proof that any behavior can be expressed via a reward function.
> > >
> > > We have changed this to "Vanilla reward learning agents do not show these behaviors", because we can see how the original sentence would be confusing.
> > >
> > > That being said, we do still stand by our intended meaning. A more formal version of the claim in Leike et al is:
> > >
> > > Every (potentially non-Markovian) deterministic policy is the uniquely optimal policy for some (potentially non-Markovian) reward function.
> > >
> > > (We say "potentially non-Markovian" because Leike et al works with histories rather than states.)
> > >
> > > In contrast, our claim is about the _learning process_, not about the optimal policy for a reward function.
> > >
> > > > By the way, you should probably cite this work
> > >
> > > Agreed, and done.
> > >
> > > Thanks for the suggestions on Figure 1 -- we will think about how to incorporate these suggestions and make it better.

---

> ### Author Response · Authors · 2020-11-24
> **It is not just about interactivity**
>
> One main claim you make is that we could get the same benefits of assistance by alternating feedback and action phases for reward learning:
>
> > Overall, I think this paper sets up a bit of a false dichotomy between (non-interactive) reward learning, and the full assistance game formulation. [...] And it is possible to achieve the same qualitative advantages by slightly modifying the (rather restrictive) formulation of reward learning that the authors use. [...] Regarding the qualitative advantages mentioned in Sections 4.1/2, we can achieve the same benefits in many instances by incorporating regular feedback sessions where H and R can communicate.
>
> Here, we argue that this does not work -- even when alternating phases in this way, you don’t get all of the qualitative benefits we have outlined. Specifically, the examples in Sections 4.1 and 4.2 would not work, and the example in Section 4.3 would work but not as well as with assistance. Intuitively the issue is that it’s not clear in this setup how R would choose when to switch between various phases, and to use beliefs about future phases to determine what to do in the current phase.
>
> Let’s consider the environment in Section 4.1 (where H is not present and will return later, and R must make a pie for her), and let’s imagine that R first optimizes expected reward under its current belief over $\theta$ for 5 timesteps (but assuming a horizon of 10 timesteps) (phase 1), then H and R can communicate (phase 2), and then R optimizes expected reward under its new belief over $\theta$ for another 5 timesteps (phase 3).
>
> In this case, R will immediately make an apple pie in phase 1, because that is the best way to get reward _under its initial belief over_ $\theta$. The key issue is that R doesn’t “know” that it should stop acting after making the dough -- that is only a good decision if it takes into account the fact that in the future it will have better information about which filling is desired. It is not clear within the reward learning paradigm how one can provide this information to R during phase 1. The assistance paradigm is the natural resolution to this issue.
>
> Now consider the environment in Section 4.2. In this case, since during phase 1, R doesn’t “know” that it can learn in the future, it will immediately pick up an apple, and if it has worms, dispose of it in an arbitrary trash can (since it never expects to know where to dispose of the apple).
>
> If you aren’t convinced, we’d be interested in a more concrete proposal for how “incorporating regular feedback sessions” would lead to the desired behavior in the examples of Section 4.1 and 4.2. We could imagine some hacks that would work for each example in particular but wouldn’t work in other situations, but we don’t want to try and argue against every possible hack that could work in this environment, and we might be missing a simple resolution that you know of.
>
> > I was confused by the claimed advantage in 4.3, since IRL already can learn from physical actions.
>
> To clarify, the advantage is that the method can learn from physical actions and can then act on its learning _within the same episode_. IRL can learn from physical actions, but it learns from actions that happened in previous episodes. We have clarified this in the paper.
>
> It is certainly possible to learn from physical actions within an episode by creating a form of interactive reward learning, for example by saying that the first 5 timesteps are for learning the reward using IRL (or something like it) and then the remaining timesteps are for acting in the environment. This would lead to similar behavior, but would require a hardcoded heuristic for choosing when to switch from the learning phase to the acting phase.

---

> > ### Comment · AnonReviewer2 · 2020-11-24
> > **The middle ground between reward learning and assistance**
> >
> > Thanks, this clarified your perspective somewhat.  I wouldn't say I'm fully convinced, however.  I think your line of argument shows that assistance has these advantages *in principle*, but this doesn't say very much about how important that is in *practice*.  We have a lot of examples in AI (e.g. neural networks) where less principled, more heuristic approaches seem to work better in practice.
> >
> > An example of something that would be in between these two paradigms would be where the agent has a distribution over possible reward functions, behaves in a risk-averse way, and queries the human when it is uncertain how to act.
> >
> > This could look something like a combination of these two papers (for example):
> > * Active Reinforcement Learning: Observing Rewards at a Cost (Krueger et al. 2016)
> > * Trial without Error: Towards Safe Reinforcement Learning via Human Intervention (Saunders et al. 2017)
> >
> > I agree with your point that you will never capture *all* of the benefits of assistance in *every* situation using something more heuristic, but I don't think that's a knock-down argument for assistance over reward learning in practice.

---

> > > ### Author Response · Authors · 2020-11-24
> > > **Agreed**
> > >
> > > >  I think your line of argument shows that assistance has these advantages in principle, but this doesn't say very much about how important that is in practice. [...] I don't think that's a knock-down argument for assistance over reward learning in practice.
> > >
> > > Yes, we agree with this, and we did not mean to imply otherwise. (Your review makes more sense now, and we can see how this was a very reasonable interpretation of what we literally wrote -- we did mention that in the future work section that we wanted to modify existing reward learning algorithms to get these benefits, even in the original submission, but that was just 1-2 sentences of the entire paper.)
> > >
> > > We have rewritten the introduction to be clearer about this. Among other edits, we have added the following paragraph:
> > >
> > > > We do not mean to suggest that all work on reward learning should cease and only research on assistive agents should be pursued. Amongst other limitations, assistive agents are very computationally complex. Our goal is simply to clarify what qualitative benefits an assistive formulation could theoretically provide. Further research is needed to develop efficient algorithms that can capture these benefits. Such algorithms may look like algorithms designed to solve assistance problems as we have formalized them here, but they may also look like modified variants of reward learning, where the modifications are designed to provide the qualitative benefits we identify.
> > >
> > > We believe that it is still an important and useful contribution to identify these qualitative behaviors, show that they can be achieved with an assistance formulation, and show that reward learning algorithms would need to be modified in order to capture the same benefits.

---

> ### Author Response · Authors · 2020-11-24
> **Human modeling is orthogonal to reward learning vs. assistance**
>
> Thanks for the review and especially the detailed suggestions on how to improve the paper! We’re glad that you found that the paper makes a nice point about the qualitative advantages of assistance.
>
> As we understand it, your main objection is that there are several advantages to the reward learning paradigm that we did not mention in the paper, leading to the paper being “one-sided”. We believe that the only major advantage of the reward learning paradigm is that it is less computationally complex (which we meant to imply in the beginning of Section 5, but weren’t sufficiently clear about; we have changed this). The other advantages that you mention seem to us to be orthogonal axes to the reward learning / assistance distinction, which we’ll discuss in more detail below.
>
> ----
>
> It seems as though you are assuming that in the assistance paradigm, it is necessary to model the human with a lot of fidelity in order to interpret all of their behavior as evidence about the reward. This is not necessary: if there is some particular domain in which we either don’t know how to interpret human behavior, or don’t want the robot to make inferences using that behavior, then we can choose an appropriately conservative human model.
>
> For example, Sections 4.1 and 4.2 use an extremely simple model, that H answers questions accurately. Even if H was capable of taking physical actions that affect the state, if we didn’t want to make any inferences based on physical actions and only wanted to rely on H’s answers to questions, we could model H as choosing physical actions uniformly at random. In this case, R can only make inferences about the reward based on H’s answers to questions. This seems to alleviate many of the concerns you raise.
>
> > it seems to require less modeling of human psychology [...] if we don't understand the semantics of H's behavior, it makes sense to restrict the communication to a more well-defined channel, as in reward learning
>
> You can choose how much modeling you want to do -- in Sections 4.1 and 4.2, if we imagine that H also has physical movement, we can choose to model question answering, but not H’s physical movement. In both paradigms, the more modeling you do, the better the results (as long as the modeling is accurate).
>
> We suspect that all three of our examples would still work if we assumed a Boltzmann-rational human model, the model used most often in reward learning.
>
> > it still seems like most effective methods are likely to lie somewhere between the two extremes described in this work.
>
> Yes, we agree that we do not want perfect modeling of $\pi^H$, especially in the near term (whether using reward learning or assistance). Our comment about better human modeling was thinking about the future, once we hit the limits of our current (very naive) modeling of $\pi^H$.
>
> > the human retains more control [...] H may not wish all of her behavior to be viewed as communicative, and allowing her to directly control when and how her behavior will be viewed as communicative grants H more agency over the behavior of R.
>
> Here also, the choice of $\pi^H$ determines which of H’s behavior is viewed as communicative. Reward learning corresponds to an assumption of some specific $\pi^H$ in a particular narrow domain, and ignorance of $\pi^H$ elsewhere; similar assumptions can be encoded with assistance.
>
> If anything, we would say that H can have more control with assistance, because H has more control over R while it is acting -- if H starts screaming “No, stop doing that!”, R will learn that its estimate of the reward is quite bad and so will stop doing whatever it is doing (see also [1]).
>
> > less opportunities to corrupt the reward signal [...] The need to more fully understand H in the assistance paradigm creates more opportunities for R to misinterpret H's behavior.
>
> Again, if you were particularly worried about this, you could design $\pi^H$ to only allow R to interpret H’s behavior in a very narrow domain.
>
> Nonetheless, we agree that in practice assistance-based solutions will rely more on extracting information from H, and so it is worse if this model is misspecified. We don’t see this as a major limitation -- it seems like this argument could be levied against nearly any method that increases the amount of information that R has about the reward, but that should simply make us more cautious about preventing misspecification, not prevent us from using the method altogether. Nonetheless, we have added a paragraph on this to the limitations section.
>
> [1] Dylan Hadfield-Menell et al. "The off-switch game." arXiv preprint arXiv:1611.08219 (2016).

---

> > ### Comment · AnonReviewer2 · 2020-11-24
> > **Regarding the framing**
> >
> > Overall, I'm still not entirely satisfied with the framing.  Assistance is a strict generalization of reward learning, as you've defined it.  But as I mentioned, "we can instead view these methods as varying wrt 1) when and how we ask R to interpret H's actions as communicative, and 2) how interactive the learning process is," with reward learning and assistance at opposite extremes.  In this case, it seems like we both agree that we might often want something in the middle, but you want to say that everything in the middle is assistance and not reward learning (which is true, as you've defined things).  But this seems to sort of ignore previous work that considers extensions of reward learning (that you would call assistance, not reward learning) to be variants of reward learning.
> >
> > Ultimately, I don't think this objection is about the naming, as much as the framing though.  I would rather the framing describe methods as being on a continuum as I've described, and discuss the pros and cons of moving in either direction along either axis of that continuum.  In this framing, my criticism of this work as "one-sided" could be rephrased as "you talk about the benefits, but not the costs, of moving towards more expressive communication channels and more interaction".  I think the benefits are important, and I expect we'll want to move increasingly in the "assistance" directions over time for the reasons you provide, but I think this paper should highlight the pros *and cons* of doing that (which it seems like you also agree are *not* merely computational).  I appreciate that you've made some updates in that direction, and I'll look at them more closely before deciding whether to maintain my score.  But I think I'd still push for a more fundamental reframing.

---

> > > ### Author Response · Authors · 2020-11-24
> > > **Communicativity and interactivity aren't the important axes of variation**
> > >
> > > We definitely want to get the framing right -- we've clearly had some trouble communicating exactly what we mean. Thanks for the help with that.
> > >
> > > > we can instead view these methods as varying wrt 1) when and how we ask R to interpret H's actions as communicative, and 2) how interactive the learning process is
> > >
> > > We agree that these are axes that methods can vary on, and that we've presented assistance and reward learning in a way that makes them look like extremes on this spectrum, but they aren't the salient axes to us. The important axis is the extent to which reward learning and control are integrated into a single decision-making process.
> > >
> > > To go into more detail, we can imagine two types of cognition -- first, "reward learning", in which the agent learns something about the unknown reward $\theta$, and second, "control", in which the agent takes actions in the environment in pursuit of reward. We call a method part of "assistance" if:
> > > 1. Decision-making for "reward learning" can take into account how that learning will be used for "control" (this enables relevance-aware questioning, as in Section 4.2), and
> > > 2. Decision-making for "control" can take into account the fact that the agent can do "reward learning" to get more information about $\theta$ (this enables plans conditional on future info, as in Section 4.1).
> > >
> > > The distinction between reward learning and assistance is primarily about whether these two types of reasoning are integrated or not. (The notion of "two phase" is trying to get at the situation where they are not integrated.) We could imagine assistance-style agents that rarely interact or communicate with humans, and we could imagine reward learning agents that have a significant amount of interaction and communication with humans.
> > >
> > > Unfortunately, our qualitative examples do tend to imply increased interactivity and communication. In some sense we would like to disavow these "extras" -- we agree that it is not clear that we always want to increase interactivity and communication. We're not really sure how to mitigate this.

---

> > > > ### Comment · AnonReviewer2 · 2020-11-24
> > > > **That was a very helpful clarification; how will you update the paper to make this clear?**
> > > >
> > > > Thanks for this clarification!
> > > > Can you sketch out (e.g. at a high level) how you might revise your submission to make this clear to the reader?

---

> > > > > ### Author Response · Authors · 2020-11-25
> > > > > **Revisions to the introduction and probably Section 4**
> > > > >
> > > > > We actually just updated the paper to rewrite the introduction (see our response in the other thread), to put more of the emphasis here. There are still more edits to make, such as changing the key insight -- while it does talk about the integration of reward learning and control, it doesn't do so very clearly. Really, we would like to put the paragraph we wrote above directly into the introduction, but we are running up against the 9-page limit. Plausibly we could rewrite the entire introduction, compressing the first three paragraphs into 1-2 sentences, as it isn't that relevant to our main point.
> > > > >
> > > > > Another potential change would be to explicitly identify what information needs to be communicated "from" reward learning "to" control or vice versa in Sections 4.1 and 4.2. This would make it clearer that this is the key feature that we are talking about.

---

> > > > > > ### Comment · AnonReviewer2 · 2020-11-25
> > > > > > **I like the planned change in sections 4.1/4.2**
> > > > > >
> > > > > > that's all.

---

> > > > ### Comment · AnonReviewer2 · 2020-11-25
> > > > **Revisiting the "reward corruption" concern**
> > > >
> > > > I think this clarification helps explain my concerns regarding assistance and reward corruption.
> > > > The core of this concern comes from this property of assistance:
> > > > > 2. Decision-making for "control" can take into account the fact that the agent can do "reward learning" to get more information about $\theta$ (this enables plans conditional on future info, as in Section 4.1).
> > > >
> > > > Naively, taking the possibility of reward learning into account when doing control seems like an invitation for reward corruption, since the agent might expect that certain actions might update its reward function in a way that would increase expected reward (directly, e.g. for any trajectory or policy).
> > > >
> > > > Such issues are discussed in these works:
> > > > * Pitfalls of learning a reward function online (Armstrong et al. 2020)
> > > > * Motivated Value Selection for Artificial Agents (Armstrong 2015).
> > > > (It looks like the 2nd of these is not available online right now, but I believe it includes the "cake or death" example, which is also discussed in this blog post, and others on LessWrong: https://www.lesswrong.com/posts/6bdb4F6Lif5AanRAd/cake-or-death)
> > > >
> > > > Can you please elaborate on whether and to what extent this is a concern for assistance and reward learning?
> > > > I also think this still deserves more discussion in the paper!

---

> > > > > ### Author Response · Authors · 2020-11-25
> > > > > **Actually, this problem is *solved* by assistance**
> > > > >
> > > > > Ah, we didn't realize you meant this sort of corruption, where we posit that the agent has an incentive to corrupt the reward learning process itself. Actually, this corruption might be a worry with interactive reward learning, but it is solved in assistance!
> > > > >
> > > > > Consider the cake-or-death problem. Quoting the linked LessWrong post:
> > > > >
> > > > > > p(C(u)|w) (probability of the correctness of u in world w) was replaced with p(C(u)|e,a) (probability of the correctness of u given the evidence e and the action a).
> > > > >
> > > > > In other words: the naive cake-or-death problem arises when the true reward is modeled as a function of the agent's observations and actions, rather than as a function of the underlying world state. But the assistance formulation explicitly models the reward as part of the state -- when we reduce the assistance game to a POMDP, the unknown reward parameters $\theta$ become part of the hidden state in the POMDP.
> > > > >
> > > > > > In summary: the sophisticated cake-or-death problem emerges for a value learning agent when it expects its utility to change predictably in certain directions dependent on its own behaviour.
> > > > >
> > > > > An optimal assistance agent can never have this. We have seen that assistance problems can be cast as POMDPs with $\theta$ as the hidden state variable. Optimal solutions to POMDPs maintain a Bayesian belief over the hidden state (in this case $\theta$). For a Bayesian belief, the expected impact of new information must be zero (conservation of expected evidence: https://www.lesswrong.com/posts/jiBFC7DcCrZjGmZnJ/conservation-of-expected-evidence ). This means that it is impossible for an optimal assistance agent to "expect [$\theta$] to change predictably in certain directions dependent on its own behaviour".
> > > > >
> > > > > This sort of problem is more possible with interactive reward learning, because you can think of these as two separate "modules" -- one that is responsible for reward learning, and one that is responsible for control. If the module that is responsible for control is "smarter" than the module responsible for reward learning, then it may be able to find behaviors that cause the reward learning module to update in a predictable way. But this crucially relies on the _separation_ between reward learning and control, which is exactly what we avoid with assistance.
> > > > >
> > > > > ----
> > > > >
> > > > > Let's turn to the pitfalls paper. It allows for arbitrary "reward learning processes", which can "update" about the reward function in arbitrary ways. However, assistance only allows for reward learning processes that are properly Bayesian (for example, it should not be possible to predict in advance how the agent will update in the future, as mentioned above).
> > > > >
> > > > > The pitfalls paper proposes a notion of uninfluencability:
> > > > >
> > > > > > Definition 2 (Uninfluenceable). The reward-function learning process $\rho$ is uninfluenceable given the prior $\xi$ on $M$ if there exists $\eta$, a probability distribution on $R$ conditional on environments, such that for all $R \in \mathcal{R}$ and $h_n \in \mathcal{H}_n$:
> > > > >
> > > > > > $P(R | h_n, \eta, \xi) = P(R | h_n, \rho)$
> > > > >
> > > > > (Here $M$ is the set of environments.) Intuitively, this criterion asks, "is there a prior $\eta$ over reward functions (given the environment), such that the future behavior of the learning process is equivalent to a Bayesian updating process starting with the prior $\eta$?"
> > > > >
> > > > > This is automatically true for the optimal policy for any assistance problem: we identify $R$ with $\theta$, $\mathcal{R}$ with $\Theta$, $M$ with $\Theta$, $\xi$ with $P_{\theta}$ and then set $\eta(\theta_{\eta} \mid \theta_{\xi}) = 1[\theta_{\eta} = \theta_{\xi}]$.
> > > > >
> > > > > In English, since the set of environments is the set of possible reward functions, if we are given a particular environment, the corresponding prior over rewards is just the reward function for that environment.
> > > > >
> > > > > The equation then holds because the optimal policy for an assistance problem can be viewed as maintaining a Bayesian belief over $\theta$, which is exactly what the criterion checks for.
> > > > >
> > > > > We are not completely confident that this math works out -- we wrote this up in the span of an hour -- but we are confident in the broader point that assistance is not subject to these sorts of corruptions.
> > > > >
> > > > > ----
> > > > >
> > > > > Overall, these "corruption" worries stem from a simple argument: if the "action-selection part" of the agent can predict how the "reward learning part" of the agent will update, then the action-selection mechanism might be incentivized to cause the reward to update in a specific way. However, assistance explicitly merges these two parts into a single unified whole, and as a result this argument no longer applies.
> > > > >
> > > > > When we were initially writing this paper, we were considering including this argument, but didn't because we expected that most readers would not care and because it seemed hard to formalize. Given that you were interested in it, maybe other readers will be as well, so we may add an appendix discussing this (if we come up with a good example environment).

---

### Official Review · AnonReviewer4 · 2020-10-29
**Official Blind Review | Reviewer #4**

**Rating:** 6
**Confidence:** 5

**Review:**

#### Summary

The submission provides a survey of two paradigms for ‘agents learning from human feedback.’ The two paradigms are unified under a new formalism (assistance games), which subsumes them as its special cases. Further, a taxonomy of different problems resulting from the formalism is provided (communicative games, two-phase games, etc.), along with illustrative examples of resulting agent behaviors. Based on the survey and taxonomy, the authors highlight that the assistance paradigm is more advantageous (in terms of possible behaviors that it can result in) than the reward learning paradigm.

=======================================================
#### Reasons for score

Strengths
  + The topic of ‘agent learning from human feedback’ is topical and of interest to the ICLR community.
  + The proposed taxonomy (i.e., assistance games) can serve as a useful common ground for discussing the different paradigms, problems, and solutions of ‘agent learning from human feedback.’
  + The paper is well written and organized.

Weaknesses
- While the submission does discuss related work (Section 2), the discussion omits several related research threads. Some of these threads have strong overlap with the setting proposed in the submission.
- Similarly, the qualitative behaviors that emerge from assistance games (Section 4) have been demonstrated in prior research (including on larger problems and with human users), thereby making it difficult to assess the novelty of the formalism.
- The key contribution of the submission is unclear (e.g., whether it is a survey, a model, a taxonomy, or all?).

I am truly on the fence regarding this submission. A novel taxonomy is certainly needed to relate and compare the diverse and growing body of research in the area of ‘agent learning from human feedback.’ However, to arrive at this taxonomy a more complete consideration of existing formalisms and algorithms is necessary. Please see suggestions listed below on prior research that relates to and, in certain cases, extends the formalism of assistance / collaboration.

=======================================================
#### Key Comments

1.	(Introduction and Proposition 1) The insight of having a single control policy for both reward learning and control modules has been previously explored. This insight is identical to that of planning / control formulations in human-robot interaction literature that model the human’s preferences (which in turn influence the reward) as latent states. These planning methods use POMDPs to represent the interaction / collaboration problem, and solve the exploration-exploitation tradeoff associated with reward learning (exploration) and control (exploitation) using POMDP solvers or MPC. Please discuss the novelty of the proposed formalism in relation to these methods. For instance,
    - (considers continuous states and actions spaces; assistance paradigm) Sadigh, Dorsa, et al. "Information gathering actions over human internal state." 2016 IEEE/RSJ International Conference on Intelligent Robots and Systems (IROS). IEEE, 2016.
    - (does not require one agent, R or H, to act before the other; assistance paradigm) Chen, Min, et al. "Planning with trust for human-robot collaboration." Proceedings of the 2018 ACM/IEEE International Conference on Human-Robot Interaction. 2018.
    - Gopalan, Nakul, and Stefanie Tellex. "Modeling and solving human-robot collaborative tasks using pomdps." RSS Workshop on Model Learning for Human-Robot Communication. 2015.
    - Nikolaidis, Stefanos, et al. "Game-theoretic modeling of human adaptation in human-robot collaboration." Proceedings of the 2017 ACM/IEEE international conference on human-robot interaction. 2017.

2. (Section 2.3) Assistance games, as defined, assume parametric specification of the hypothesis space of the reward / preference of humans. However, nonparametric extensions (both for assistance and reward learning) have been proposed. Please consider relating the proposed formalism with these prior works. For instance,
  - Michini, Bernard, and Jonathan P. How. "Bayesian nonparametric inverse reinforcement learning." Joint European conference on machine learning and knowledge discovery in databases. Springer, Berlin, Heidelberg, 2012.
  - Panella, Alessandro, and Piotr Gmytrasiewicz. "Bayesian learning of other agents' finite controllers for interactive POMDPs." Proceedings of the Thirtieth AAAI Conference on Artificial Intelligence. 2016.

3. (Section 2.3) In assistance games, as defined, the reward (or human preferences over reward) does not change during the task. However, extensions exist which model the latent state corresponding to reward as being locally active and / or time varying. Please consider relating the proposed formalism with these related works. For example,
  - (locally active reward; reward learning paradigm) Michini, Bernard, and Jonathan P. How. "Bayesian nonparametric inverse reinforcement learning." Joint European conference on machine learning and knowledge discovery in databases. Springer, Berlin, Heidelberg, 2012.
  - (locally active reward; reward learning paradigm) Park, Daehyung, et al. "Inferring Task Goals and Constraints using Bayesian Nonparametric Inverse Reinforcement Learning." Conference on Robot Learning. PMLR, 2020.
  - (considers time varying preferences with learned dynamics; assistance paradigm) Nikolaidis, Stefanos, David Hsu, and Siddhartha Srinivasa. "Human-robot mutual adaptation in collaborative tasks: Models and experiments." The International Journal of Robotics Research 36.5-7 (2017): 618-634.
  - (considers time varying preferences with learned dynamics; assistance paradigm) Unhelkar, Vaibhav V., Shen Li, and Julie A. Shah. "Semi-Supervised Learning of Decision-Making Models for Human-Robot Collaboration." Conference on Robot Learning. 2020.

4.  The behaviors arising from solving assistance games (Section 4) have been previously formalized by multiple human-AI collaboration approaches and demonstrated with human users. For instance,
  -	(exhibits behaviors outlined in Section 4.2) Kamar, Ece, Ya’akov Gal, and Barbara J. Grosz. "Incorporating helpful behavior into collaborative planning." Proceedings of The 8th International Conference on Autonomous Agents and Multiagent Systems (AAMAS). Springer Verlag, 2009.
  -	(reasons about communicative actions) Whitney, David, et al. "Reducing errors in object-fetching interactions through social feedback." 2017 IEEE International Conference on Robotics and Automation (ICRA). IEEE, 2017.
  -	(reasons about both physical actions and communications) Nikolaidis, Stefanos, et al. "Planning with verbal communication for human-robot collaboration." ACM Transactions on Human-Robot Interaction (THRI) 7.3 (2018): 1-21.
  -	(reasons about both physical actions and communications) Unhelkar, Vaibhav V., Shen Li, and Julie A. Shah. "Decision-Making for Bidirectional Communication in Sequential Human-Robot Collaborative Tasks." Proceedings of the 2020 ACM/IEEE International Conference on Human-Robot Interaction. 2020.
  -	(communicative actions) Liang, Claire, et al. "Implicit communication of actionable information in human-ai teams." Proceedings of the 2019 CHI Conference on Human Factors in Computing Systems. 2019.

Please discuss the connection of the proposed formalism with these approaches.

5.	(Section 2.2) Please also consider discussing the following related works on active reward learning,
  - Lopes, Manuel, Francisco Melo, and Luis Montesano. "Active learning for reward estimation in inverse reinforcement learning." Joint European Conference on Machine Learning and Knowledge Discovery in Databases. Springer, Berlin, Heidelberg, 2009.
  - Brown, Daniel S., Yuchen Cui, and Scott Niekum. "Risk-aware active inverse reinforcement learning." Conference on Robot Learning. 2018.
  - Tschiatschek, Sebastian, et al. "Learner-aware teaching: Inverse reinforcement learning with preferences and constraints." Advances in Neural Information Processing Systems. 2019.
  - Cui, Yuchen, and Scott Niekum. "Active reward learning from critiques." 2018 IEEE International Conference on Robotics and Automation (ICRA). IEEE, 2018.

=======================================================
#### Minor Comment

- (Section 2.1) Please consider including a note explaining the asterisk notation (used to define the domain for POMDP policy).

=======================================================

---

> ### Author Response · Authors · 2020-11-24
> **We have incorporated the works you mention**
>
> Thanks for the review, and especially for the very detailed elaboration of related works!
>
> > The key contribution of the submission is unclear (e.g., whether it is a survey, a model, a taxonomy, or all?)
>
> Unfortunately this isn’t a standard “type” of paper. Our paper compares and contrasts the assistance paradigm to the (very common) reward learning paradigm, making it neither a model, nor a survey, nor a taxonomy.
>
>
> > the qualitative behaviors that emerge from assistance games (Section 4) have been demonstrated in prior research [...], thereby making it difficult to assess the novelty of the formalism.
>
> As far as we can tell, while some qualitative behaviors have been shown by prior work, they use very different formalisms. For example, amongst your citations, the results of Section 4.2 have only been shown using probabilistic recipe trees (PRTs), which are _extremely_ different from the assistance paradigm.
>
> We do agree that the behavior in Section 4.3 has been demonstrated previously (we cited shared autonomy). We included it anyway because it is an important contrast to reward learning. We were considering whether to add a Section 4.4 in which R can use its physical actions to “query” H, but decided against it, precisely because this had been shown in much prior work.
>
> > While the submission does discuss related work (Section 2), the discussion omits several related research threads.
>
> We are unsure whether you are viewing this as a survey paper -- many of the research threads you list are certainly related, but delving into all of the related threads would be a paper in and of itself.
>
> > (Introduction and Proposition 1) The insight of having a single control policy for both reward learning and control modules has been previously explored.
>
> Yes, we agree this has been explored before -- we cited some of these works in Section 2.3.1. On the specific papers you suggested:
>
> > Information gathering actions over human internal state
>
> While this is not an exact fit for the active reward learning formalism, since the robot’s “queries” do affect the state of the environment, it is much closer to that than to full assistance. The robot chooses its actions to maximize information gain, not value of information. This means that we don’t get the benefits of Section 4. In particular, in Section 4.1, the method would function like interactive reward learning -- it would wait until H arrived, then ask about their preferences, and only then start to make the pie. In Section 4.2, the method would either immediately ask where to dispose of wormy apples (if the information gain is worth the time cost) or would never ask (if the information gain is too small to be worth it).
>
> > Planning with trust for human-robot collaboration
>
> Thanks, this is a good example of the use of an assistance-style formalism. We have added a sentence to Section 2.3, alongside Macindoe et al (2012) and Woodward et al (2019). (Note though that this paper does not discuss or demonstrate any of the qualitative behaviors we identify, nor does it compare against reward learning.)
>
> > Modeling and solving human-robot collaborative tasks using pomdps
>
> This is another good example.
>
> > Game-theoretic modeling of human adaptation in human-robot collaboration
>
> This paper seems to be about how the robot can teach the human, rather than about how the human can teach the robot as in our setting. A quote: “we assume that in the beginning of the game, the robot has perfect information about the reward matrix”.
>
> > Assistance games, as defined, assume parametric specification of the hypothesis space of the reward / preference of humans. However, nonparametric extensions (both for assistance and reward learning) have been proposed.
>
> The parametric definition is simply for exposition; our results also apply to the non-parametric case. We value exposition over exhaustiveness, since our focus is on conceptual clarity
>
> > (Section 2.3) In assistance games, as defined, the reward (or human preferences over reward) does not change during the task. However, extensions exist which model the latent state corresponding to reward as being locally active and / or time varying. Please consider relating the proposed formalism with these related works.
>
> A detailed discussion is out of scope, as it isn’t relevant to the meat of the paper. We have added a very brief discussion to Section 5.1.
>
> > The behaviors arising from solving assistance games (Section 4) have been previously formalized by multiple human-AI collaboration approaches and demonstrated with human users.
>
> Thanks for the citations -- we have added details to Sections 4.2 and 4.3. (None of these seem related to Section 4.1.)
>
> > Please also consider discussing the following related works on active reward learning,
>
> The listed papers are central examples of active reward learning, which we discussed (with citations) in Section 2.2. It is beyond our scope to discuss each specific paper.

---

> > ### Comment · AnonReviewer4 · 2020-11-24
> > **Thank you for the detailed response.**
> >
> > Thanks for clarifying that the paper isn't of a "standard type." Papers certainly don't need to be. However, my concern about the core contributions still remain.
> >
> > As highlighted in the reviews and response, there have been several assistance or assistance-style formalisms in prior work which exhibit the behaviors described in Section 4. Given this, I take the key messages of the paper as:
> > 1. reward learning is a special case of assistance games (as defined), and
> > 2. given 1, assistance games can exhibit additional behaviors that reward learning cannot, and
> > 3. given 2, assistance games should be used to generate agent behavior.
> >
> > However, to achieve these benefits, the assistance paradigm also imposes additional requirements on the problem setting (e.g., it requires presence of both human and robot in the same episode). In particular, there are several settings where reward learning is possible and assistance is not, making the third message moot for these settings.

---

> > > ### Author Response · Authors · 2020-11-24
> > > **We maintain that the behaviors in Section 4.1 and 4.2 are novel**
> > >
> > > > there have been several assistance or assistance-style formalisms in prior work which exhibit the behaviors described in Section 4.
> > >
> > > We continue to disagree with this, in the case of Sections 4.1 and 4.2 -- it seems to us that:
> > > - Prior work has shown the qualitative behavior of Section 4.2 using probabilistic recipe trees
> > > - Prior work has created assistance-based algorithms, and demonstrated them on environments that don't lead to the qualitative behaviors of Sections 4.1 and 4.2
> > > - No prior work has connected the behaviors of Sections 4.1 and 4.2 to the assistance formalism.
> > >
> > > In your review, you mention the following papers:
> > >
> > > > (exhibits behaviors outlined in Section 4.2) Kamar, Ece, Ya’akov Gal, and Barbara J. Grosz. "Incorporating helpful behavior into collaborative planning." Proceedings of The 8th International Conference on Autonomous Agents and Multiagent Systems (AAMAS). Springer Verlag, 2009.
> > >
> > > This is the one that uses probabilistic recipe trees.
> > >
> > > > (reasons about communicative actions) Whitney, David, et al. "Reducing errors in object-fetching interactions through social feedback." 2017 IEEE International Conference on Robotics and Automation (ICRA). IEEE, 2017.
> > > > (reasons about both physical actions and communications) Nikolaidis, Stefanos, et al. "Planning with verbal communication for human-robot collaboration." ACM Transactions on Human-Robot Interaction (THRI) 7.3 (2018): 1-21.
> > > > (reasons about both physical actions and communications) Unhelkar, Vaibhav V., Shen Li, and Julie A. Shah. "Decision-Making for Bidirectional Communication in Sequential Human-Robot Collaborative Tasks." Proceedings of the 2020 ACM/IEEE International Conference on Human-Robot Interaction. 2020.
> > > > (communicative actions) Liang, Claire, et al. "Implicit communication of actionable information in human-ai teams." Proceedings of the 2019 CHI Conference on Human Factors in Computing Systems. 2019.
> > >
> > > These are all about treating physical actions as communicative, and so are examples of the qualitative behavior of Section 4.3, but not Sections 4.1 and 4.2.
> > >
> > > > In particular, there are several settings where reward learning is possible and assistance is not
> > >
> > > Taken literally this cannot be true, since reward learning is a strict subset of assistance. We assume you are claiming that a "full assistance approach" where we have complete interactivity and mixing of learning and acting phases is impossible. This may be the case in some settings, but there are also settings where assistance is feasible (as with, e.g., future household robots), and if we can do so it would be worth getting the qualitative benefits of assistance in these situations.

---

### Official Review · AnonReviewer5 · 2020-11-08
**The paper provides a comparative analysis of reward learning and assistance. While the discussion in the paper is certainly interesting, I'm uncertain about what are the contributions of the paper.**

**Rating:** 5
**Confidence:** 3

**Review:**

### Overview

The paper addresses the problem of learning from human feedback. It provides an analysis of reward learning---where human feedback is used to extract a task description in the form of a reward---and assistance---where the learning agent and human co-exist in the environment and both perform actions; the agent seeks to select its actions to optimize the (unknown) that is implicit in the human's actions.

The paper shows that reward learning problems can be converted to assistance problems, turning queries from reward learning to communicative actions in a two-phase communicative assistance problem. Conversely, two-phase communicative assistance problems can be converted to active reward learning problems.

### Comments

In my opinion, the paper reads very well, and the discussion in the paper is quite interesting.

In spite of an interesting discussion, I am not certain about the contribution of the paper. The derivations, although they seem technically sound, are not particularly surprising; similarly, the qualitative differences in behavior pointed out in the paper are hardly surprising---assistance learning, being a more general problem than reward learning (as follows from the discussion of the paper), will lead to more diverse behaviors.

The particular behaviors observed are also a consequence of the POMDP formulation, as the agent will act to strike an adequate tradeoff between information gathering and goal optimization.

### Post-rebuttal update

I thank the authors for the clarifications and discussion. However, and admitting that I may have missed something, I remain unconvinced regarding the contributions of the paper.

---

> ### Author Response · Authors · 2020-11-24
> **The contribution is in identifying the qualitative behaviors and restrictions**
>
> Thanks for the review! We’re glad that you found the discussion interesting and the paper well-written.
>
> > I am not certain about the contribution of the paper.
>
> To recap, the primary contributions of our paper are:
> 1. Identifying reward learning problems as special cases of assistance problems; in particular corresponding to assistance problems that are _communicative_ and _two phase_.
> 2. Illustrating the qualitative behaviors that can arise when we remove these restrictions on assistance problems.
>
> > The derivations, although they seem technically sound, are not particularly surprising
>
> We think there is value in precisely defining the two characteristics of reward learning (communicative and two phase) that distinguish it from assistance, and showing that these are exhaustive (i.e. there are no other differences that we have failed to uncover). We do not think the theorems were particularly technically challenging or surprising once the two characteristics were identified.
>
> > assistance learning, being a more general problem than reward learning (as follows from the discussion of the paper), will lead to more diverse behaviors.
>
> It is certainly not surprising that by generalizing the problem setup we are able to get new behaviors, and we do not claim that it is. Our novel contribution is to illustrate what exactly these new behaviors are, and why they might be particularly useful. As an analogy, multiagent environments are a generalization of single agent environments, but it is still useful to study what these new behaviors are (cooperation, competition, commitments, bargaining, etc).
>
> It also seems to us that the other reviewers did not find the qualitative behaviors particularly obvious.
>
> > The particular behaviors observed are also a consequence of the POMDP formulation, as the agent will act to strike an adequate tradeoff between information gathering and goal optimization.
>
> Analogously, all theorems are consequences of their axioms. But surely after stating a set of axioms, there is still useful work to be done in proving theorems?
>
> We assume you meant that the key feature of POMDPs is that they allow for trading off information gathering and goal optimization (which was known before our work), and all of the behaviors we outline are easily-understood consequences of that fact. We do expect researchers who have spent a lot of time working on human-robot interaction in POMDPs to intuitively understand most of these qualitative behaviors, but we would not expect a typical AI researcher to identify these qualitative behaviors given just the formalisms for them.

---

### Decision · Program_Chairs · 2021-01-07
**Final Decision**

**Decision:**

Reject

**Comment:**

This is a well written paper, outlining a class of assistive algorithms. Being more or less a survey paper, it could do a better job of discussing 'inverse reinforcement learning' and 'collaborative inverse reinforcement learning'. It could also be slightly more general: for example the human dewcision function need not be known if we model the interaction as a Bayesian game (then the human might have a latent type, which can be inferred together with the reward function). The active reward learning problem is sometimes referred to as 'preference elicitation'. In the end, it was not clear that the discussion in this paper had any actionable insights for future models or algorithms in this area.